# In-Token Learning for High-Fidelity Image Restoration via Diffusion Transformers

## Abstract

Diffusion-based image restoration has advanced rapidly, yet existing methods remain fragile under severe degradations, exhibiting geometric drift, identity loss, or texture hallucination. We present **In-Token Learning**, a token-aligned framework that redefines restoration as learning a conditional velocity field via rectified flow matching (RFM), directly transporting pure noise to clean images under intra-token alignment within a Multimodal Diffusion Transformer (MMDiT). This design enables robust and high-fidelity restoration, avoiding misleading details from degraded inputs. To further stabilize conditioning, we introduce **Direct Low-Quality Guidance (DLG)**, a lightweight mechanism that injects degraded-image and prompt embeddings into model's native text-conditioning pathway, without relying on external prompts, side branches, or sequence-level concatenation.

Our framework (i) improves robustness under severe degradations, (ii) improves fidelity by narrowing the long-standing perception-distortion gap, and (iii) supports QHD ($2560\times1440$) inference and seamless scaling to ultra-high resolutions through fixed-length attention. We further demonstrate the first 12K restoration of the classical scroll painting *Along the River During the Qingming Festival* using an unmodified backbone. Across five benchmarks (DIV2K, LSDIR, FFHQ, RealLQ250, RealPhoto60), our method achieves state-of-the-art performance on both full- and no-reference metrics, and generalizes to colorization, achieving state-of-the-art perceptual quality. These results position In-Token Learning as a unified and scalable paradigm across diverse tasks, degradations, and resolutions.

## 1 Introduction

Recent advances in diffusion models Ho et al. (2020); Rombach et al. (2022); Podell et al. (2023) have significantly improved image restoration quality, especially in super-resolution (SR) and real-world deblurring. Diffusion Transformers (DiTs) Peebles & Xie (2023); Esser et al. (2024a) combine strong generative priors with long-range attention, enabling high-fidelity detail synthesis. However, despite their perceptual strength, two key limitations remain.

First, current diffusion-based restoration systems Wang et al. (2024); Lin et al. (2024) often fail under severe degradations such as strong noise, motion blur, or low-resolution compression. These conditions frequently lead to geometric drift, identity loss, or texture hallucination—especially when training is restricted to $\leq 1024^2$ pixels Yu et al. (2024); Ai et al. (2024). Second, although no-reference perceptual scores (e.g., CLIPIQA Wang et al. (2023), MUSIQ Ke et al. (2021), MANIQA Yang et al. (2022)) are often high, full-reference distortion metrics (e.g., PSNR, SSIM) remain low—commonly referred to as the perception-distortion gap Blau & Michaeli (2018). Moreover, high-resolution inference is limited by quadratic attention and tight architecture coupling, making tile-consistent scaling difficult without modification.

We address these challenges by proposing **In-Token Learning**. Our token-aligned framework redefines diffusion-based image restoration by replacing iterative denoising of degraded inputs with direct noise-to-clean mapping via a conditional velocity field learned with rectified flow matching (RFM) Liu et al. (2022). Unlike prompt- or control-based methods Yu et al. (2024); Chen et al. (2025), or in-context sequence-level concatenation Huang et al. (2024); Labs et al. (2025), our framework performs restoration at the token level: evolving latent tokens are fused with degraded-input tokens along the channel dimension. This design preserves fixed-length attention, making the

framework resolution-agnostic. Moreover, this design separates artifacts in degraded inputs from the latent representation, making our framework robust even under severe degradations. As a result, our framework not only preserves structure and identity but also inherently supports high-resolution QHD ($2560\times1440$) inference and scales seamlessly to 4K, 8K, and even 12K resolution with tile-consistent scaling. To further enhance learning stability and semantic conditioning, we introduce a lightweight mechanism: **Direct Low-Quality Guidance (DLG)**. DLG injects a fused embedding of the degraded image and a per-task system prompt into the text-conditioning pathway. Unlike prior works (e.g., SUPIR Yu et al. (2024), FaithDiff Chen et al. (2025)) that rely on external vision-language models (VLMs) Liu et al. (2023) or ControlNet-style side branches Zhang et al. (2023), DLG provides compact, task-aware guidance at minimal cost.

On super-resolution and denoising, we validate our approach on five benchmarks spanning synthetic (DIV2K Agustsson & Timofte (2017), LSDIR Li et al. (2023), FFHQ Karras et al. (2019)) and real-world (RealLQ250 Ai et al. (2024), RealPhoto60 Yu et al. (2024); Chen et al. (2025)). Our method achieves state-of-the-art results on RealPhoto60 and all synthetic datasets (DIV2K, LSDIR, FFHQ). Beyond super-resolution, our framework also generalizes to automatic colorization, achieving state-of-the-art fidelity and perceptual quality with conservative, material-consistent chroma.

**Contributions.**

- **In-Token Learning:** A new paradigm for image restoration that learns a conditional velocity field via rectified flow matching (RFM), enabling direct noise-to-clean generation under in-token alignment and DLG, in contrast to prior approaches that iteratively denoise degraded inputs.

- **Direct Low-Quality Guidance:** A lightweight guidance mechanism that injects task-aware information by fusing degraded-image and prompt embeddings through the native text-conditioning pathway, without external VLMs or ControlNet-style branches.

- **Narrowing the perception-distortion gap:** Our approach consistently improves both full-reference and perceptual metrics across multiple benchmarks and degradation settings.

- **Ultra-high resolution:** Benefiting from the fixed-length attention of in-token alignment, our framework supports direct QHD inference and tile-consistent 4K/8K/12K inference. We demonstrate this by successfully restoring a 12K classical scroll painting.

- **Task generalization:** The same backbone and training pipeline extend beyond super-resolution to colorization, demonstrating strong cross-task transfer without redesign.

## 2 RELATED WORK

### 2.1 EARLY METHODS FOR IMAGE RESTORATION

Early image restoration methods (e.g., BSRGAN Zhang et al. (2021), Real-ESRGAN Wang et al. (2021)) are trained as deterministic feed-forward regressors with pixel- or perceptual-loss objectives. While achieving relatively high scores in full-reference metrics, they under-model the conditional target distribution. Consequently, under severe degradations or large upscales, they often exhibit structural distortions or identity drift, leading to degraded no-reference perceptual quality.

### 2.2 DIFFUSION AND DiT FOR RESTORATION

Diffusion priors Ho et al. (2020) significantly improve perceptual quality in super-resolution and other restoration tasks Saharia et al. (2023); Lin et al. (2024). Evolving from U-Nets Ronneberger et al. (2015), DiT Peebles & Xie (2023); Esser et al. (2024a) variants enable global attention by tokenizing latent features. Although perceptual quality improves, the perception-distortion gap persists, limiting fidelity under severe degradations Blau & Michaeli (2018); Deng et al. (2025). Moreover, they are typically limited to training resolutions $\leq 1024^2$ pixels, restricting scalability.

### 2.3 CONDITIONING AND TOKEN ALIGNMENT

Preserving structural and identity information in diffusion models Ho et al. (2020) remains challenging due to their iterative denoising nature. Many existing works Yu et al. (2024); Ai et al. (2024);

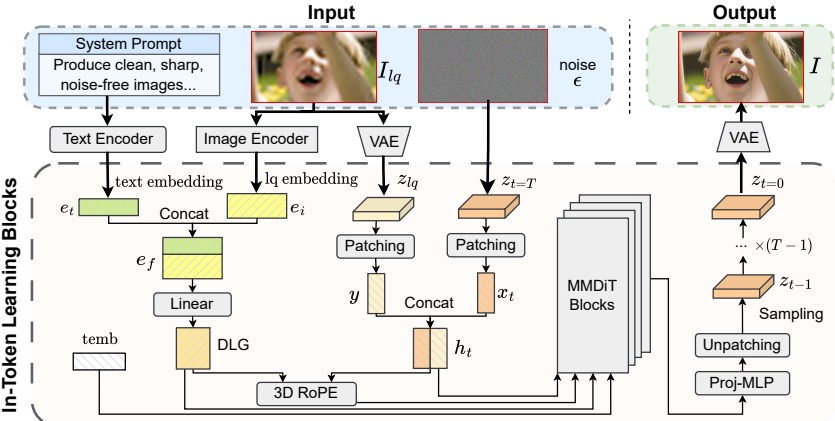

Figure 1: **In-Token Learning overview.** Our framework learns a conditional velocity field that transports noise toward the clean image, conditioned by in-token alignment and DLG. This intra-token design improves robustness under severe degradations, narrows the perception-distortion gap, supports direct QHD and tile-consistent 4k/8k/12k inference.

Chen et al. (2025); Deng et al. (2025) leverage external VLMs Liu et al. (2023) to obtain semantic cues from degraded inputs, while also incorporating ControlNet-like side branches Zhang et al. (2023) to enforce structural consistency. However, they are constrained by a denoising-fidelity trade-off and additional computational overhead, while still failing to bridge the perception-distortion gap.

More recently, in-context concatenation approaches Huang et al. (2024); Labs et al. (2025) attempt to align degraded inputs with denoising latents by expanding token sequences. While helpful, sequence-level fusion inflates attention cost quadratically and complicates high-resolution scaling.

## 3 METHOD

### 3.1 OVERVIEW

We introduce a new image restoration paradigm that directly map pure noise to clean image via RFM, conditioned by in-token alignment and DLG. Instead of denoising degraded latents with generic diffusion priors, we explicitly define a degradation model $\mathcal{D}_\phi$ to synthesize paired training data. Our approach is then trained to learn a conditional velocity field that maps noise to the clean latent, conditioned by the degraded input generated by this degradation model.

As shown in Fig. 1, our method generates restored images from pure noise during inference. At each step $t$, noisy-latent tokens $x_t$ and low-quality tokens $y$ (from degraded image input) are concatenated channel-wise into $h_t = [x_t; y]$, which is processed by MMDiT blocks. In parallel, DLG injects a concatenation fusion of the degraded image input and a task prompt into the text-conditioning pathway. Together, explicit degradation modeling, in-token alignment, and lightweight semantic guidance yield robust structure/identity fidelity without auxiliary ControlNets or external VLMs.

### 3.2 DEGRADATION MODEL AND SYNTHETIC SUPERVISION

Given a clean image $I$, $\mathcal{D}_\phi$ degrades it into a low-quality counterpart $I_{lq}$. This process enables supervised restoration training by defining the ground-truth velocity field from pure noise to $I$, conditioned on $I_{lq}$. For super-resolution and denoising,

$$I_{lq} = \mathcal{D}_\phi(I) = \mathrm{JPEG}_q\big((I * \kappa_{\sigma_b}) \downarrow_s \big) + \eta, \quad \eta \sim \mathcal{N}(0, \sigma_n^2), \tag{1}$$

where $s$ is the downsampling factor, $\kappa_{\sigma_b}$ is a Gaussian blur kernel with standard deviation $\sigma_b$, $q$ denotes JPEG quality, and $\eta$ is additive Gaussian noise with standard deviation $\sigma_n$. For colorization, $\mathcal{D}_\phi$ converts RGB images to grayscale:

$$I_{lq} = \mathcal{D}_\phi(I) = \mathcal{G}(I), \tag{2}$$

where $\mathcal{G}$ denotes grayscale conversion. This unified formulation ensures controllable coverage, and enables systematic stress-testing across degradations.

### 3.3 NOISE-TO-CLEAN MAPPING VIA RECTIFIED FLOW

Let $E, D$ denote the encoder and decoder of a pretrained VAE Rombach et al. (2022). Given $(I_{\text{lq}}, I = \mathcal{D}_\phi^{-1}(I_{\text{lq}}))$, we obtain clean and degraded latents $z_0 = E(\mathcal{D}_\phi^{-1}(I_{\text{lq}}))$ and $z_{\text{lq}} = E(I_{\text{lq}})$.

We formulate restoration as learning a conditional velocity field Liu et al. (2022) $v_\theta$ that transports a noisy latent

$$z_t = (1 - t)z_0 + t\epsilon, \quad \epsilon \sim \mathcal{N}(0, \mathrm{I}), \tag{3}$$

towards the clean latent $z_0$. This direct flow-based formulation avoids propagating artifacts from $z_{\text{lq}}$, unlike prior approaches that iteratively denoise degraded latents. Conditioning is injected at two complementary levels: spatial alignment via in-token fusion and semantic guidance via DLG.

### 3.4 RECTIFIED FLOW MATCHING OBJECTIVE

We adopt rectified flow matching Lipman et al. (2022) to train the model. The training loss is:

$$\mathcal{L}_{\text{RFM}} = \mathbb{E}_{z_0, \epsilon, t}\Big[\ \|v_\theta(z_t, t, e_f, [x_t;\ E(I_{lq})]) - (\epsilon - E(\mathcal{D}_\phi^{-1}(I_{\text{lq}})))\|_2^2\ \Big]. \tag{4}$$

This provides direct supervision on the transport direction, offering stable convergence compared with score-based objectives. More theoretical notes are provided in Appendix A.

### 3.5 IN-TOKEN ALIGNMENT

To efficiently inject degraded features, we concatenate latent tokens $x_t$ and $y$ along the channel dimension to form in-token fusion:

$$h_t = [x_t; y] \in \mathbb{R}^{N \times 2d}, \tag{5}$$

with $N$ tokens and hidden size $d$. As shown in Fig. 1, $y$ is concatenated to the channel dimension of $x_t$ at each denoising step. This design greatly improves restoration fidelity while doubling only the embedding dimension and keeping attention cost manageable; see Section 3.8 for detailed analysis.

### 3.6 DIRECT LOW-QUALITY GUIDANCE

To let our proposed method learn the reverse process of the degradation model more directly, we use a system prompt to provide semantic guidance for the reverse process. We also calculate the low-quality embedding $e_i$ from $I_{\text{lq}}$ via a frozen image encoder Labs (2024b). We provide the final guidance by fusing the fixed system prompt embedding $e_t$ with degraded low-quality embedding $e_i$:

$$e_f = [e_t; e_i] \in \mathbb{R}^{(n_t + n_i) \times c}, \tag{6}$$

where $n_t, n_i$ are token counts and $c$ the channel width. $e_f$ is fed into the model's text-conditioning pathway at every step. Unlike VLM-based captioning or ControlNet-style branches, DLG is lightweight, task-aware, and introduces no extra modules.

### 3.7 INFERENCE DETAILS

For resolutions up to QHD ($2560 \times 1440$), each denoising step is performed in a single forward pass. For ultra-high resolutions (4K/8K/12K), we tile both $z_{\text{lq}}$ and the degraded image $I_{\text{lq}}$. The tiled $I_{\text{lq}}$ is then used to obtain the corresponding embeddings $e_{i'}$, while $z_{\text{lq}}$ provides $z_{\text{lq}'}$. The same inference paradigm is applied consistently to each tile, ensuring tile-consistent restoration without architectural modification. Details are provided in Appendix B.

### 3.8 COMPLEXITY AND SCALABILITY

Self-attention complexity scales quadratically with sequence length. In-context concatenation doubles the sequence to $2N$, yielding $\mathcal{O}((2N)^2 d)$ cost. Our in-token fusion preserves length $N$ and doubles only the channel width, reducing cost to $\mathcal{O}(N^2 \cdot 2d)$. This halves FLOPs and quarters memory usage relative to sequence-level fusion, enabling training directly at QHD resolution and efficient tiled 4K/8K/12K inference. See Appendix C for a formal derivation.

## 4 EXPERIMENTS

We evaluate In-Token Learning primarily on super-resolution and denoising, and further test automatic image colorization to demonstrate cross-task generalization. To highlight scalability, we also showcase the first 12K restoration demo of *Along the River During the Qingming Festival*, generated with our super-resolution and denoising model. Due to space and resolution constraints, the complete 12K outputs and video demo are provided in the anonymous repository.

### 4.1 SETUP

**Test Datasets.** For synthetic SR and denoising, we use a subset of FFHQ Karras et al. (2019) and the validation sets of DIV2K Agustsson & Timofte (2017), LSDIR Li et al. (2023) with three degradation levels (D1/D2/D3: $\times 2/ \times 4/ \times 8$ downsampling with increasing blur, noise, and JPEG compression); For real-world SR, we use RealLQ250 Ai et al. (2024) and RealPhoto60 Yu et al. (2024); Chen et al. (2025). For automatic colorization, different from prior works which mainly test on ImageNet Deng et al. (2009) at $256^2$ resolution, we convert the validation sets of DIV2K and LSDIR to grayscale images and evaluate at their original resolution.

**Implementation Details.** We build on FluxFill Labs (2024a) by disabling the mask mechanism and adopting our in-token paradigm. We fine-tune the DiT at bf16 precision with LoRA Hu et al. (2022) (rank 384) for 14k steps on $4 \times$RTX 5880 Ada, using a cosine learning-rate schedule with peak $2.5 \times 10^{-4}$, 2.5k warm-up steps, a global batch size of 16, and 28 sampling steps at inference.

**Metrics.** For super-resolution and denoising, we report full-reference (PSNR/SSIM/LPIPS) and no-reference (MANIQA Yang et al. (2022), CLIPIQA Wang et al. (2023), MUSIQ Ke et al. (2021)) metrics, following prior works Yu et al. (2024); Chen et al. (2025). For automatic image colorization, we include the colorfulness score (CF) Hasler & Suesstrunk (2003), and $\Delta$CF, the absolute deviation between the output CF and the ground-truth CF, following DDColor Kang et al. (2023).

### 4.2 SUPER-RESOLUTION AND DENOISING

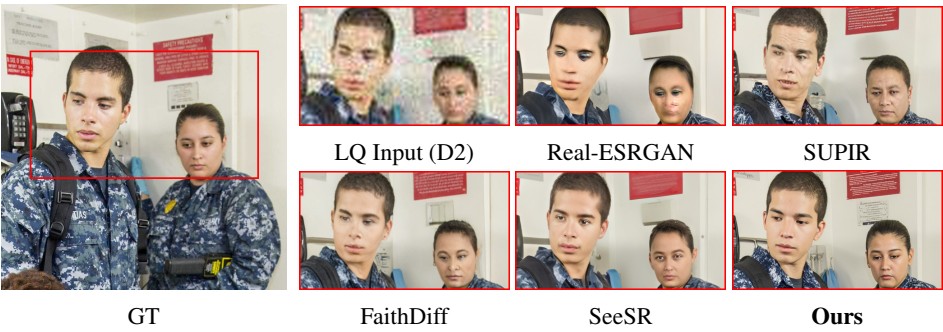

| LQ Input (D2) | Real-ESRGAN | SUPIR |

| GT | FaithDiff | SeeSR | **Ours** |

Figure 2: **Qualitative comparisons on *DIV2K-Val* (synthetic).** Instead of denoising from a degraded latent, our method generates from pure noise using a conditional velocity field guided by in-token alignment and DLG. As a result, it produces reconstructions with highest fidelity that better preserve eye shape, gaze direction, thin edges, and high-frequency textures.

**Quantitative.** Table 1 reports results on DIV2K, LSDIR, FFHQ, RealLQ250, and RealPhoto60, against GAN baseline (Real-ESRGAN Wang et al. (2021)) and recent diffusion-based methods (StableSR Wang et al. (2024), DiffBIR Lin et al. (2024), SeeSR Wu et al. (2024), SUPIR Yu et al. (2024), DreamClear Ai et al. (2024), FaithDiff Chen et al. (2025)).

On synthetic datasets, our approach consistently ranks first or second across both full-reference (PSNR/SSIM) and no-reference (CLIPIQA/MUSIQ/MANIQA) metrics, effectively narrowing the perception-distortion gap that commonly hinders diffusion-based restorers. While Real-ESRGAN often attains the highest distortion-oriented PSNR/SSIM scores, such metrics correlate imperfectly with human perception; our higher no-reference scores and qualitative results (Figs. 2, 3) demonstrate superior perceptual fidelity with fewer artifacts.

Table 1: Quantitative comparison on the validation set of *DIV2K*, *LSDIR*, *FFHQ* (synthetic), *RealLQ250*, and *RealPhoto60* (real-world). For synthetic datasets, no-reference metrics (CLIP-IQA/MUSIQ/MANIQA) are averaged over D1–D3. Full-reference metrics (PSNR/SSIM) are reported for D1 only. For real-world sets, we only apply $4\times$ upscale on *RealLQ250* and $2\times$ upscale on *RealPhoto60*. Best is **bold+underline** and second-best is **bold**.

| Datasets | Metrics | Methods | | | | | | | |
|----------|---------|------------|----------|---------|--------|-------|-----------|----------|------|
| | | Real-ESRGAN | StableSR | DiffBIR | SeeSR | SUPIR | DreamClear | FaithDiff | **Ours** |
| *DIV2K-Val* | CLIPIQA↑ | 0.4875 | 0.3801 | 0.5737 | **0.6035** | 0.5051 | 0.5310 | 0.5763 | **0.6213** |
| | MUSIQ↑ | 58.9699 | 49.0826 | 66.1573 | 69.0464 | 64.6142 | 65.9251 | **68.1122** | 66.3780 |
| | MANIQA↑ | 0.5252 | 0.4444 | 0.5709 | 0.6015 | 0.5858 | 0.5743 | **0.6184** | **0.6218** |
| | PSNR↑ | **26.0669** | 24.1885 | **25.3999** | 24.5178 | 25.3269 | 23.3814 | 24.3723 | 25.2706 |
| | SSIM↑ | 0.7678 | 0.7044 | 0.7036 | 0.6738 | 0.7025 | 0.6414 | 0.6609 | 0.7057 |
| | LPIPS↓ | 0.2997 | 0.3186 | 0.3226 | 0.3434 | 0.3107 | 0.3464 | 0.3302 | 0.3087 |
| *LSDIR-Val* | CLIPIQA↑ | 0.5424 | 0.4199 | 0.6309 | **0.6413** | 0.5704 | 0.6121 | 0.6307 | **0.7217** |
| | MUSIQ↑ | 64.5601 | 52.8770 | 70.1140 | 72.6831 | 67.9508 | 70.3541 | 71.6307 | **72.6592** |
| | MANIQA↑ | 0.5674 | 0.4788 | 0.6153 | 0.6388 | 0.6234 | 0.6149 | **0.6653** | **0.6880** |
| | PSNR↑ | 23.2872 | 21.3458 | 22.3082 | 21.5545 | 22.2406 | 21.0995 | 21.1642 | **22.7338** |
| | SSIM↑ | 0.7220 | 0.6241 | 0.6355 | 0.5859 | 0.6431 | 0.6029 | 0.5702 | **0.6809** |
| | LPIPS↓ | 0.2805 | 0.3164 | 0.3052 | 0.3349 | 0.3017 | 0.3190 | 0.3255 | **0.2819** |
| *FFHQ* | CLIPIQA↑ | 0.4297 | 0.4827 | 0.6283 | 0.5493 | 0.5300 | 0.4807 | 0.5719 | **0.5779** |
| | MUSIQ↑ | 63.4199 | 66.7731 | 73.9354 | 73.3842 | 73.6046 | 70.7586 | 76.1941 | **74.1305** |
| | MANIQA↑ | 0.4920 | 0.5029 | 0.5925 | 0.5875 | 0.5970 | 0.5654 | 0.6406 | **0.6151** |
| | PSNR↑ | 30.9152 | 28.8965 | 29.8403 | 29.4371 | 30.2578 | 28.2804 | 28.6319 | **30.3368** |
| | SSIM↑ | 0.8448 | 0.7911 | 0.7866 | 0.7851 | 0.7901 | 0.7467 | 0.7434 | **0.7932** |
| | LPIPS↓ | 0.3196 | 0.2981 | 0.3426 | 0.3181 | **0.2991** | 0.3309 | 0.3201 | 0.3162 |
| *RealLQ250* | CLIPIQA↑ | 0.4586 | 0.3554 | 0.5410 | 0.5668 | 0.4862 | 0.5085 | **0.5466** | 0.4792 |
| | MUSIQ↑ | 57.4012 | 46.1955 | 62.7923 | 67.1609 | 60.9576 | 61.8146 | **66.5082** | 54.3427 |
| | MANIQA↑ | 0.5333 | 0.4578 | 0.5883 | **0.6026** | 0.5816 | 0.5749 | 0.6298 | 0.5397 |
| *RealPhoto60* | CLIPIQA↑ | 0.5068 | 0.3632 | 0.5465 | 0.5959 | **0.5972** | 0.5383 | 0.5718 | **0.6028** |
| | MUSIQ↑ | 59.0296 | 50.2670 | 61.6646 | 71.8052 | 69.6326 | 68.8353 | **71.5853** | 65.7139 |
| | MANIQA↑ | 0.4797 | 0.4554 | 0.5617 | 0.6079 | 0.6116 | 0.5905 | 0.6513 | **0.6168** |

On real-world benchmarks, our method sets a new state of the art on RealPhoto60. On RealLQ250, while aggregate scores are slightly lower, a closer analysis (Fig. 3) shows that for the majority of images whose degradations fall inside our degradation model (downsampling, blur, Gaussian noise, JPEG compression), our approach consistently surpasses all baselines in both perceptual quality and structural fidelity. In long-tail cases with degradations outside the scope of our degradation model, our framework adopts a reliability-oriented strategy: instead of hallucinating implausible content, it preserves faithful geometry and identity. This behavior reflects a favorable trade-off: delivering stronger performance where reliable restoration is feasible, while avoiding misleading artifacts under high uncertainty. Such conservatism is advantageous in high-stakes scenarios requiring trustworthy restoration, such as the restoration of historical artifacts and artworks (e.g., low-resolution or degraded paintings and scrolls).

**Qualitative.** Our approach achieves the highest fidelity and perceptual quality both on synthetic (Fig. 2) and real-world (Fig. 3) datasets. Existing diffusion-based methods often struggle to balance strong denoising or large upscale with faithful structure/identity preservation, occasionally introducing geometric drift or texture hallucination under severe degradations. Our approach produces cleaner reconstructions with tighter alignment to input geometry and materials, retaining fine structures (e.g., eye shape, gaze direction, thin edges, high-frequency textures) while avoiding oversmoothing and aggressive hallucination. We provide detailed visualizations in Appendix D including per-degradation level comparisons (Fig. 7), results on DIV2K (Fig. 8) and RealPhoto60 (Fig. 9), and additional failure cases on RealLQ250 (Appendix 10), where our method preserves input structures under uncertain degradations while competing approaches hallucinate implausible content.

### 4.3 GENERALIZATION TO AUTOMATIC IMAGE COLORIZATION

**Baselines and protocol.** We compare to four recent automatic colorization systems with public code and checkpoints: DDColor Kang et al. (2023), BigColor Kim et al. (2022), ColorFormer Ji et al. (2022), and InstColor Su et al. (2020). Some baselines enforce a low fixed test resolution

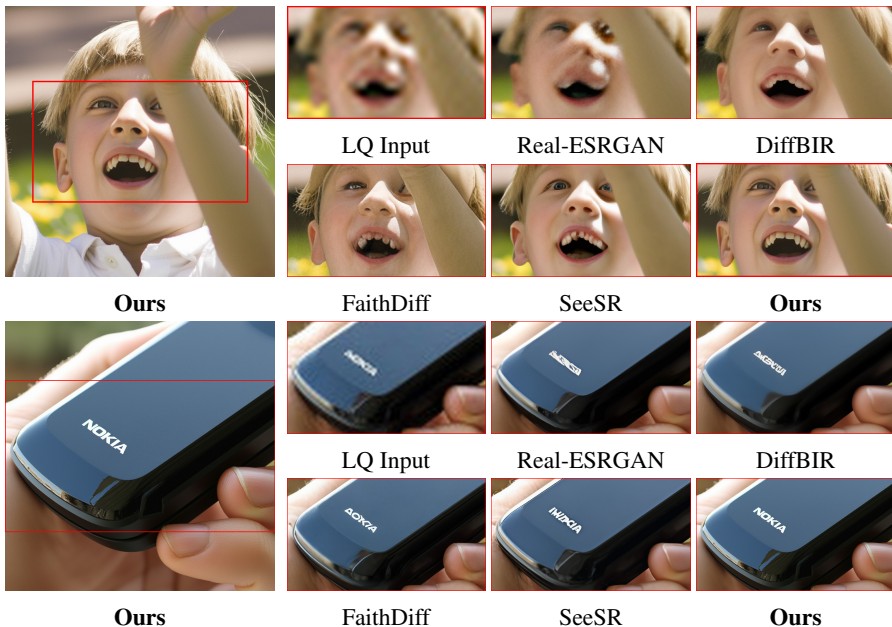

Figure 3: **Qualitative comparisons on *RealLQ250* (real-world).** Left: our full-resolution results with crop locations marked (red boxes). Right: low-quality input and outputs from Real-ESRGAN, DiffBIR, FaithDiff, SeeSR, and our method on the corresponding crops. Our approach achieves the highest fidelity and perceptual quality on real-world datasets.

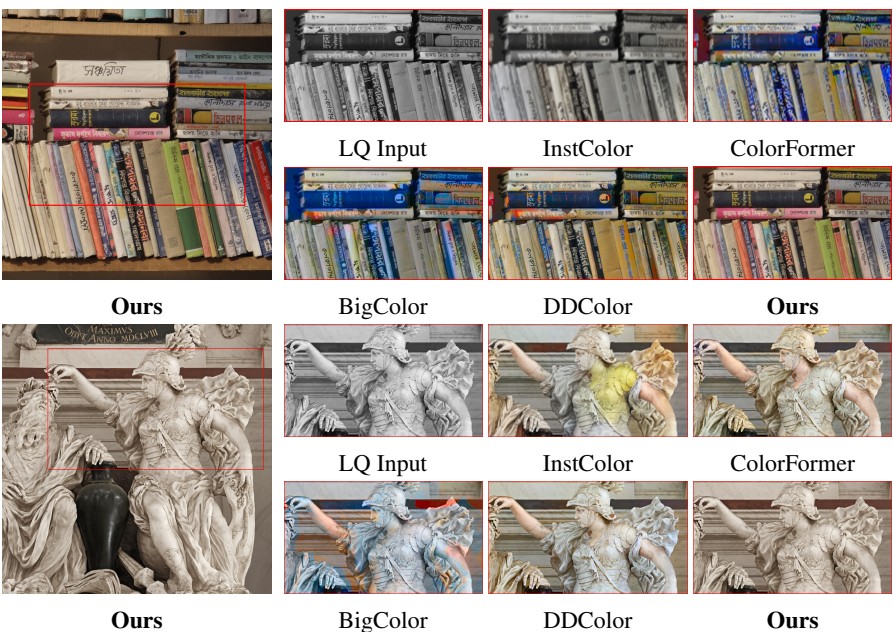

Figure 4: **Qualitative comparisons on *DIV2K-Val* (colorization).** Left: our full-resolution result with crop locations marked (red boxes). Right: grayscale input and outputs from InstColor, ColorFormer, BigColor, DDColor, and our method on the corresponding crops. Our method produces most material-consistent colors while preserving luminance and fine details.

(e.g., $256 \times 256$ in ColorFormer Ji et al. (2022) and InstColor Su et al. (2020)). To reduce resolution-induced bias while respecting their codebases, we run those models at their maximum stable resolution ($512 \times 512$) and bicubic-resize back to the input size for evaluation.

Table 2: **Colorization on *DIV2K-Val* and *LSDIR-Val* (sRGB).** We report CF (↑), ΔCF (↓), PSNR (↑), and three no-reference perceptual metrics: MANIQA, CLIPIQA, and MUSIQ (↑). Our method attains the best PSNR and the best perceptual scores on both datasets, despite using more restrained colorfulness (lower CF). Best is **bold+underline** and second best is **bold**.

| Method | DIV2K-Val | | | | | | LSDIR-Val | | | | | |
|---|---|---|---|---|---|---|---|---|---|---|---|---|
| | CF↑ | ΔCF↓ | PSNR↑ | MANIQA↑ | CLIPIQA↑ | MUSIQ↑ | CF↑ | ΔCF↓ | PSNR↑ | MANIQA↑ | CLIPIQA↑ | MUSIQ↑ |
| *InstColor* | 31.4139 | 19.8368 | 20.8960 | 0.3886 | 0.3184 | 31.9294 | 31.5915 | 30.0520 | 19.4124 | 0.5315 | 0.4110 | 51.2842 |
| *ColorFormer* | 42.5202 | **12.9178** | 20.6107 | 0.4668 | 0.3703 | 39.3247 | **44.0540** | **21.9420** | 19.3988 | 0.5758 | 0.4849 | 58.0788 |
| *BigColor* | **43.0503** | 14.6197 | 20.4527 | 0.5532 | 0.4231 | 62.8257 | 43.6812 | 25.6883 | 19.3395 | 0.6251 | 0.5389 | 70.7619 |
| *DDColor* | **48.8789** | 14.0947 | 21.9010 | 0.5551 | 0.4481 | 64.1891 | **53.9194** | **17.9724** | 20.5454 | 0.6318 | 0.5651 | 71.4752 |
| **Ours** | 29.3404 | 19.0102 | **22.8351** | **0.6137** | **0.5276** | **64.4435** | 37.6489 | 25.0217 | **20.8218** | **0.6838** | **0.6467** | **71.7749** |

**Quantitative.** Table 2 shows that our approach achieves the highest PSNR and consistently outperforms prior methods on all three no-reference metrics (MANIQA, CLIPIQA, MUSIQ) across both datasets. Although some baselines report higher CF or smaller ΔCF, these results are often associated with over-saturation or artifacts, which are penalized by perceptual metrics. In contrast, our model produces more conservative and materially consistent chroma, preserving luminance structures and yielding higher perceptual fidelity.

**Qualitative.** Figure 4 shows visual comparisons on DIV2K-Val. Our method generates realistic, material-consistent colors while preserving grayscale textures and lighting. Other approaches may introduce over-saturated tones or unnatural hues (e.g., bluish casts on book spines or skin-tone tints on marble statues), which can compromise global coherence. The zoomed regions highlight our better detail fidelity, restrained saturation, and more coherent scene appearance. Additional examples are provided in Appendix 12.

## 4.4 ABLATION STUDIES

All ablations are conducted on *DIV2K-Val* at the D2 level in the SR&denoising setting, under strictly matched training/inference budgets and fixed random seeds. Overall, the full model with pure noise generation, in-token alignment, and DLG with both *Text Emb.* ($e_t$) and *LQ Emb.* ($e_i$), offers the best results both on full-reference and no-reference metrics.

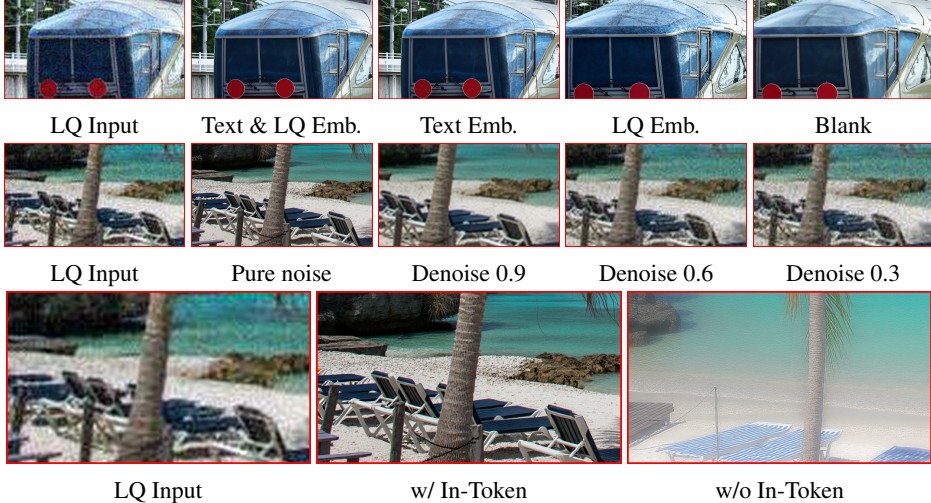

| LQ Input | Text & LQ Emb. | Text Emb. | LQ Emb. | Blank |

| LQ Input | Pure noise | Denoise 0.9 | Denoise 0.6 | Denoise 0.3 |

| LQ Input | w/ In-Token | w/o In-Token |

Figure 5: **Ablations for SR & denoising.** *Top:* DLG variants show that combining text and LQ embeddings (Text & LQ Emb.) yields the best structure and detail. *Middle:* starting from pure noise performs better than denoising-initialization at varying strengths (0.9/0.6/0.3). *Bottom:* removing **In-Token** alignment (w/o In-Token) degrades both structure and perceptual quality.

**Ablation of DLG.** We decompose DLG into *Text Emb.* (global task prior) and *LQ Emb.* (instance evidence) and evaluate the 2×2 combinations (Fig. 5, Tab. 3). **Blank** yields blurry, incomplete

Table 3: **DLG ablation on *DIV2K-Val* (D2).** We toggle *Text Emb.* and *LQ Emb.* in DLG.

| Text Emb. | LQ Emb. | PSNR↑ | SSIM↑ | CLIPIQA↑ |
|:---:|:---:|:---:|:---:|:---:|
| ✗ | ✗ | 15.5284 | 0.3811 | 0.5102 |
| ✓ | ✗ | 16.2637 | 0.4271 | 0.5623 |
| ✗ | ✓ | 15.0445 | 0.3408 | 0.5984 |
| ✓ | ✓ | **21.9795** | **0.5698** | **0.6293** |

Table 4: **Generation regime on *DIV2K-Val* (D2).** Pure noise vs. denoising initialization.

| Pure noise | Denoising | PSNR↑ | SSIM↑ | CLIPIQA↑ |
|:---:|:---:|:---:|:---:|:---:|
| ✗ | 0.30 | 16.0334 | 0.3218 | 0.2072 |
| ✗ | 0.60 | 16.2681 | 0.3565 | 0.2240 |
| ✗ | 0.90 | 16.4119 | 0.4432 | 0.3633 |
| ✓ | — | **21.9795** | **0.5698** | **0.6293** |

Table 5: **Effect of in-token alignment on *DIV2K-Val* (D2).** Enabling alignment substantially improves both full-reference (PSNR/SSIM) and no-reference(CLIPIQA/MUSIQ/MANIQA) metrics.

| In-Token Alignment | PSNR↑ | SSIM↑ | CLIPIQA↑ | MUSIQ↑ | MANIQA↑ |
|:---:|:---:|:---:|:---:|:---:|:---:|
| ✗ | 11.8003 | 0.2932 | 0.4538 | 47.1899 | 0.5417 |
| ✓ | **21.9795** | **0.5698** | **0.6293** | **66.8804** | **0.6176** |

results (the train wiper disappears), hurting both full-/no-reference metrics. **LQ only** is sharper but "dirty": noise in the LQ input bends the wiper and suppresses PSNR/SSIM, even as perceptual quality may rise (e.g., CLIPIQA $0.510{\rightarrow}0.598$). **Text only** is clean yet over-smoothed; roof seams and the wiper remain implausible. Slightly improve full-/no-reference metrics. **DLG (Text+LQ)** uses the text prior to constrain and denoise LQ evidence, anchoring structure while filtering artifacts; both full-/no-reference metrics improve (e.g., PSNR $21.98\,\text{dB}$), producing the best results.

**Ablation of Generation Regime: pure noise vs. denoising.** Our in-token design allows generating the restored image directly from pure noise, bypassing the classical "denoise the LQ" route. We vary denoising strength $\{0.30, 0.60, 0.90\}$ while keeping the schedule fixed (Fig. 5, Tab. 4). Even with strong denoising (0.90), anchoring to flawed LQ inputs depresses perceptual quality (CLIP-IQA 0.363 vs. 0.629 for pure noise) and leaves a large PSNR gap (16.41 vs. $21.98\,\text{dB}$). Increasing $s$ moderately lifts PSNR/SSIM within the denoising family, but never closes the gap to pure noise generation and consistently hurts no-reference metrics—suggesting that early reliance on corrupted evidence propagates artifacts.

**Ablation of In-Token Alignment.** Finally, removing token-level alignment during training and inference causes a substantial drop in fidelity and structural coherence, degrading both full-reference and no-reference metrics (Fig. 5, Tab. 5). With alignment enabled, the model preserves the fidelity, underscoring that token-aligned supervision is central to our approach.

## 5 CONCLUSION

We introduce In-Token Learning, a new paradigm for general-purpose image restoration that achieves high-resolution and high-fidelity across diverse tasks. In contrast to prior approaches that rely on diffusion priors and iteratively denoise degraded latents, our framework learns a conditional velocity field from pure noise to clean images conditioned by in-token alignment and DLG. Our framework improves structural and identity fidelity, narrows the perception-distortion gap, and scales seamlessly from QHD to ultra-high 12K resolution.

Looking forward, enriching the degradation model to better cover diverse long-tail cases and additional restoration tasks offers a path toward greater generalization. With this broader perspective, In-Token Learning stands as a scalable and extensible foundation for next-generation high-fidelity restoration systems, bridging research prototypes and practical deployment at ultra-high resolutions.

## USE OF LLMS

We used a large language model (ChatGPT) only for language polishing (rephrasing and grammar correction). All ideas, experiments, and claims are the authors' own, and we take full responsibility for the content.

REPRODUCIBILITY STATEMENT

The 12K restoration demo on *Along the River During the Qingming Festival* is available via an anonymous link: `https://drive.google.com/uc?export=download&id=1DYE9smvb_BPPinJgEOGcLXriVH494Sep`. We also provide anonymized inference code and pretrained checkpoints in an anonymous repository: `https://anonymous.4open.science/r/E06BDFD2-6126-4F62-99FC-927BF0840D35`. The accompanying README includes side-by-side wipe animations for super-resolution and colorization, demonstrating the fidelity of our method at ultra-high resolution.

ETHICS STATEMENT

This work uses only publicly available datasets obtained from open Internet sources, and does not involve any private or sensitive data.

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

USE OF LLMS

We used a large language model (ChatGPT) only for language polishing (rephrasing and grammar correction). All ideas, experiments, and claims are the authors' own, and we take full responsibility for the content.

# A  ADDITIONAL THEORETICAL NOTES

## A.1  RECTIFIED FLOW MATCHING

Our In-Token Learning framework is trained via rectified flow matching Lipman et al. (2022); Liu et al. (2022); Esser et al. (2024b), which formulates image restoration as learning a time-dependent velocity field that deterministically transports noise to the clean data distribution in latent space.

**Forward Process.** Let $x_0 \sim p_{\text{data}}$ be the clean target latent and $\epsilon \sim \mathcal{N}(0, \mathrm{I})$ Gaussian noise. Rectified flow defines a linear interpolation path between $x_0$ and $\epsilon$:

$$z_t = (1 - t)x_0 + t\epsilon, \quad t \in [0, 1], \tag{7}$$

where $z_0 = x_0$ and $z_1 = \epsilon$. This deterministic path simplifies the generative process into an ODE trajectory in latent space Lipman et al. (2022).

**Velocity Field and Training Target.** The model $v_\theta(z_t, t, h_t, e_f)$ predicts the velocity that transports $z_t$ towards $x_0$. The ground-truth velocity is:

$$\frac{d}{dt}z_t = \epsilon - x_0. \tag{8}$$

Hence, the training objective is the $\ell_2$ regression:

$$\mathcal{L}_\theta = \mathbb{E}_{t, z_0, \epsilon, I_{\text{lq}}}\Big[\|v_\theta(z_t, t, h_t, e_f) - (\epsilon - x_0)\|_2^2\Big]. \tag{9}$$

**Conditioning with In-Token Learning.** Our model conditions the velocity field on two sources of information: (1) the in-token fusion $h_t = [x_t; y]$, where degraded tokens $y$ are concatenated with noisy latent tokens $x_t$ along the channel dimension; and (2) the fused embedding $e_f$ of the system prompt and low-quality image introduced by DLG. Formally, the conditional velocity predictor is defined as

$$v_\theta(z_t, t, h_t, e_f) \approx v^*(z_t, t), \tag{10}$$

where $v^*(z_t, t)$ denotes the target velocity under rectified flow matching. This formulation ensures that structural cues from degraded tokens and semantic cues from prompts remain consistently integrated across timesteps.

## A.2  IN-TOKEN CONDITIONING WITH ROPE

We use Rotary Positional Embeddings (RoPE) Su et al. (2024) in spatial form $(h, w)$ for both $x_t$ and $y$, since channel-wise fusion keeps the sequence length unchanged:

$$h_t = [x_t; y] \in \mathbb{R}^{N \times 2d}. \tag{11}$$

This preserves the spatial grid size while temporal information is still provided by the standard timestep embedding. As a result, the Transformer can naturally attend to location-wise correspondences, while channel fusion provides redundancy for detail preservation.

## A.3  TRAINING DYNAMICS

The rectified flow loss in Eq. 9 yields gradients:

$$\nabla_\theta \mathcal{L}_\theta = \mathbb{E}_{t, z_0, \epsilon, I_{\text{lq}}}\Big[2\big(v_\theta(z_t, t, h_t, e_f) - (\epsilon - x_0)\big)\nabla_\theta v_\theta(z_t, t, h_t, e_f)\Big]. \tag{12}$$

Intuitively: - Small $t$ emphasizes fine detail reconstruction from $x_0$; - Large $t$ encourages noise suppression and structural guidance from $y$.

This dynamic naturally complements our design: early steps benefit from degraded-image tokens $y$ to anchor structure, while later steps leverage $e_f$ for semantic consistency.

## B INFERENCE DETAILS

### B.1 DIRECT QHD INFERENCE

For resolutions up to QHD ($2560 \times 1440$), we run an ODE sampler on full images with the same in-token fusion at each step.

### B.2 TILE-CONSISTENT ULTRA-HIGH RESOLUTION INFERENCE

We detail our tiled inference scheme, which enables seamless 4K/8K/12K restoration without architectural modifications. For higher resolutions we tile $z_{\mathrm{lq}}$ in latent space and tile $I_{\mathrm{lq}}$ in image space to obtain the corresponding tiled embeddings $e_i^{(\mathrm{tile})}$. At each step $t$: (1) split $z_t$ and $z_{\mathrm{lq}}$ into overlapping crops (stride $s$, overlap $o$), yielding $\{z_t^{(\mathrm{tile})}, z_{\mathrm{lq}}^{(\mathrm{tile})}\}$; (2) crop $I_{\mathrm{lq}}$ accordingly to compute $e_i^{(\mathrm{tile})}$; (3) build a fused embedding $e_f^{(\mathrm{tile})} = [e_t; e_i^{(\mathrm{tile})}]$, where $e_t$ is the global system prompt shared across tiles; (4) patch $z_t^{(\mathrm{tile})}$ and $z_{\mathrm{lq}}^{(\mathrm{tile})}$ into tokens $\{x_t^{(\mathrm{tile})}, y^{(\mathrm{tile})}\}$, forming $h_t^{(\mathrm{tile})} = [x_t^{(\mathrm{tile})}; y^{(\mathrm{tile})}]$; (5) update each crop with one denoising step and blend them with smooth overlaps to assemble the global latent $\tilde{z}_{t-\Delta t}$.

After reaching $t=0$, we decode with $D(\cdot)$. This hybrid guidance—global $e_t$ and tile-local $e_i^{(\mathrm{tile})}$—keeps semantics consistent while grounding each tile locally, yielding seam-free boundaries.

## C COMPLEXITY ANALYSIS

Let an $H \times W$ input produce $N = \frac{H \times W}{p^2}$ image tokens (patch size $p$), channel dimension $d$, $h$ heads, $d_k$ per-head dimension, $n_f = (n_t + n_i) \ll N$ prompt tokens, and $L$ Transformer layers. The dominant cost/memory of attention is quadratic in the sequence length.

### C.1 IN-CONTEXT VS. IN-TOKEN CONDITIONING

We compare the FLOPs and memory scaling of sequence-level and channel-level fusion.

**In-context visual conditioning (sequence concat).** The degraded image is appended as another $N$ tokens (plus $n_f$ prompt tokens), giving length $2N+n_f$:

$$\mathrm{FLOPs}_{\mathrm{ctx}} \propto L\,h\,(2N+n_f)^2\,d_k, \qquad \mathrm{Mem}_{\mathrm{ctx}} \propto (2N+n_f)^2.$$

**In-token learning (channel concat).** We keep the sequence at $N+n_f$ (image tokens plus a short prompt stream) and only widen channels by $2\times$; attention cost scales with width linearly:

$$\mathrm{FLOPs}_{\mathrm{token}} \propto L\,h\,(N+n_f)^2\,(2d_k), \qquad \mathrm{Mem}_{\mathrm{token}} \propto (N+n_f)^2.$$

### C.2 IMPLICATIONS FOR SCALABILITY

For $n_f \ll N$, the ratio simplifies to

$$\frac{\mathrm{FLOPs}_{\mathrm{ctx}}}{\mathrm{FLOPs}_{\mathrm{token}}} \approx \frac{(2N)^2\,d_k}{N^2\,(2d_k)} = \mathbf{2}\times,$$

and the attention-memory ratio is $\approx \frac{(2N)^2}{N^2} = \mathbf{4}\times$.

This constant-factor efficiency gap is key to enabling direct training at QHD resolution and practical inference at 4K/8K/12K, making In-Token Learning scalable without model surgery.

## D ADDITIONAL ANALYSES AND RESULTS

**Inference precision.** We study the quality–efficiency trade-off at inference time by comparing **bf16** and **qfloat8** Hugging Face Optimum Team (2025). For each precision, we report restoration

quality (PSNR/SSIM, CLIPIQA, MUSIQ, MANIQA) together with the peak GPU memory requirement (VRAM) measured at **QHD (1440p)** inference. On *DIV2K-Val* (D2), qfloat8 reduces peak VRAM by ∼43% (30→17 GB) but consistently lowers quality across all metrics (Tab. 7). This clarifies when ultra-low precision is attractive—memory-bound scenarios—and when bf16 is preferable for quality-critical use cases.

**SuperResolution&denoising.** We first show the detailed per-level quantitative result, showing more baselines. We then show more visualizations of the validation set of DIV2K on degradation level D3. We therefore show case the low quality input at different level of degradation, and give a vivide comparison of the methods ability when degradation increases. Finally, we give more detailed visualizations on RealPhoto60, showing our methods ability.

**Colorization.** We present extended side-by-side visual comparisons against recent methods. Our approach produces sharper structures and more faithful colors while avoiding common artifacts such as over-saturation and hue shifts.

Table 6: **Efficiency comparison with recent diffusion-based restoration models.** Our method achieves the **lowest trainable parameter count** while maintaining competitive performance. Inference time is measured at $1536^2$ resolution on $2\times$ RTX 5880 Ada GPUs.

| Method | SeeSR | SUPIR | DreamClear | FaithDiff | Ours |
|---|---|---|---|---|---|
| Trainable Params (B) ↓ | 2.3 | 1.3 | 2.41 | 2.4 | **1.0** |
| Inference Speed (s, $1536^2$) ↓ | 34 | **26** | 107 | 35 | 75 |

Table 7: **Inference precision ablation on *DIV2K-Val* (D2).** At **QHD (1440p)** inference, switching from **bf16** to **qfloat8** cuts the *peak GPU memory* from 30 to 17 GB ($\approx$43%↓), but yields lower generation quality across fidelity (PSNR/SSIM) and perceptual metrics (CLIPIQA, MUSIQ, MANIQA).

| Precision | PSNR↑ | SSIM↑ | CLIPIQA↑ | MUSIQ↑ | MANIQA↑ | VRAM (GB)↓ |
|---|---|---|---|---|---|---|
| qfloat8 | 18.5053 | 0.4707 | 0.5096 | 63.8563 | 0.5886 | **17** |
| **bf16** | **21.9795** | **0.5698** | **0.6293** | **66.8804** | **0.6176** | 30 |

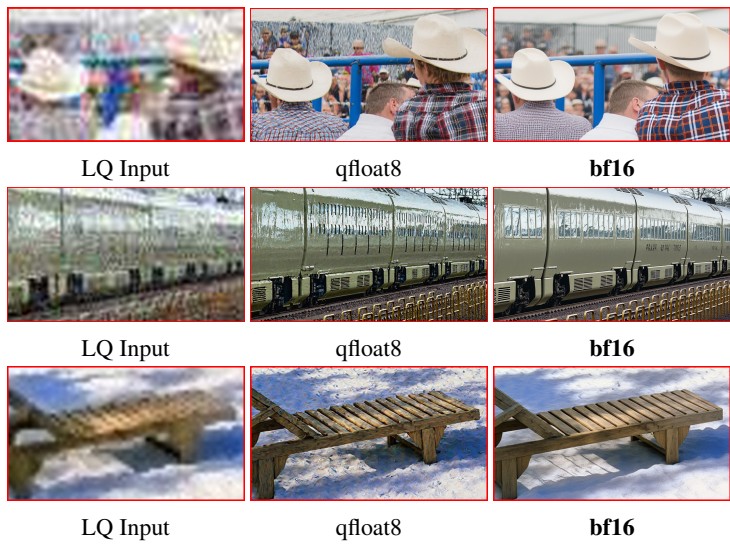

|  |  |  |
|---|---|---|
| LQ Input | qfloat8 | **bf16** |
| LQ Input | qfloat8 | **bf16** |
| LQ Input | qfloat8 | **bf16** |

Figure 6: **Qualitative comparison under different inference precisions (qfloat8 vs. bf16).** Each triplet shows the low-quality input, our model inferred with **qfloat8**, and with **bf16**. At **QHD (1440p)** inference, qfloat8 lowers the *peak* VRAM from **30** to **17** GB, but under challenging settings, it more often suppresses fine structures and introduces spurious "dirty" textures.

Table 8: Results on *DIV2K_VAL*, *LSDIR_VAL*, and *FFHQ-face* across D1–D3. Higher is better for CLIPIQA/MUSIQ/MANIQA/PSNR/SSIM (↑); lower is better for LPIPS (↓). Best is **bold+underline** and second best is **bold**.

| Datasets | D-Level | Metrics | BSRGAN | Real-ESRGAN | SwinIR | DASR | StableSR | DiffBIR | SeeSR | SUPIR | DreamClear | FaithDiff | Ours |
|---|---|---|---|---|---|---|---|---|---|---|---|---|---|
| *DIV2K-Val* | D1 | CLIPIQA↑ | 0.4930 | 0.5785 | 0.3581 | 0.5663 | 0.5048 | **0.6254** | **0.6101** | 0.5186 | 0.5830 | 0.5755 | 0.5745 |
| | | MUSIQ↑ | 65.1796 | 67.7866 | 48.5841 | 67.8606 | 65.5961 | **69.3433** | 69.3333 | 65.5327 | 68.1176 | 68.0778 | 64.7165 |
| | | MANIQA↑ | 0.5734 | 0.5870 | 0.5586 | 0.5705 | 0.5949 | 0.5921 | 0.6028 | 0.6061 | 0.5946 | **0.6173** | **0.6111** |
| | | PSNR↑ | **27.9713** | 26.0669 | **26.3220** | 26.2952 | 24.1885 | 25.4000 | 24.5178 | 25.3269 | 23.3814 | 24.3723 | 25.2706 |
| | | SSIM↑ | **0.8068** | 0.7678 | 0.6824 | **0.7798** | 0.7044 | 0.7036 | 0.6738 | 0.7025 | 0.6414 | 0.6609 | 0.7057 |
| | | LPIPS↓ | **0.2832** | **0.2997** | 0.4493 | 0.3183 | 0.3186 | 0.3226 | 0.3434 | 0.3107 | 0.3464 | 0.3302 | 0.3088 |
| | D2 | CLIPIQA↑ | 0.5025 | 0.4757 | 0.3277 | 0.4155 | 0.3734 | 0.5681 | **0.5948** | 0.5111 | 0.5260 | 0.5814 | **0.6293** |
| | | MUSIQ↑ | 64.0924 | 62.0101 | 33.9770 | 59.9372 | 53.6485 | 66.0401 | **68.8320** | 65.2617 | 66.2470 | **68.5392** | 66.8804 |
| | | MANIQA↑ | 0.5062 | 0.5234 | 0.3397 | 0.4489 | 0.4811 | 0.5977 | 0.5977 | 0.5926 | 0.5752 | **0.6219** | **0.6176** |
| | | PSNR↑ | 23.9390 | **23.2248** | 22.7615 | 23.1437 | 23.6248 | 23.8260 | 23.1446 | 22.7877 | 22.1618 | 22.7779 | 21.9795 |
| | | SSIM↑ | **0.6354** | 0.6308 | 0.5007 | 0.6210 | 0.6303 | 0.6173 | 0.5998 | 0.5791 | 0.5701 | 0.5692 | 0.5698 |
| | | LPIPS↓ | 0.4107 | 0.4020 | 0.5879 | 0.4258 | 0.3976 | 0.4144 | 0.4043 | 0.3994 | 0.4042 | **0.3959** | **0.3925** |
| | D3 | CLIPIQA↑ | 0.4024 | 0.4084 | 0.2420 | 0.3492 | 0.2621 | 0.5278 | **0.6056** | 0.4856 | 0.4840 | 0.5722 | **0.6602** |
| | | MUSIQ↑ | 44.7879 | 47.1131 | 32.5216 | 41.5056 | 28.0033 | 63.0886 | **68.9738** | 63.0481 | 63.4108 | **67.7197** | 67.5373 |
| | | MANIQA↑ | 0.4196 | 0.4652 | 0.1834 | 0.3984 | 0.2573 | 0.5594 | **0.6041** | 0.5587 | 0.5532 | 0.6161 | **0.6368** |
| | | PSNR↑ | **21.3738** | 20.7643 | 20.1131 | 20.6904 | 21.0975 | **21.2616** | 20.1085 | 19.8682 | 19.5761 | 20.0535 | 19.0548 |
| | | SSIM↑ | **0.5296** | **0.5255** | 0.4187 | 0.5117 | 0.4908 | 0.4985 | 0.4684 | 0.4427 | 0.4556 | 0.4420 | 0.4541 |
| | | LPIPS↓ | 0.5117 | 0.5031 | 0.6663 | 0.5202 | 0.5977 | 0.5132 | 0.5018 | 0.4993 | 0.4953 | **0.4892** | **0.4737** |
| *LSDIR-Val* | D1 | CLIPIQA↑ | 0.5443 | 0.6408 | 0.4231 | 0.6303 | 0.5645 | **0.6759** | 0.6496 | 0.6139 | 0.6674 | 0.6432 | **0.7112** |
| | | MUSIQ↑ | 69.8149 | 72.9823 | 58.1202 | **73.0371** | 69.7710 | **73.0705** | 73.0090 | 71.7411 | 72.3086 | 72.0226 | 72.7672 |
| | | MANIQA↑ | 0.6232 | 0.6538 | 0.6003 | 0.6360 | 0.6392 | 0.6497 | 0.6451 | 0.6700 | 0.6478 | **0.6721** | **0.6886** |
| | | PSNR↑ | **24.7521** | 23.2872 | **23.6155** | 23.5496 | 21.3458 | 22.3082 | 21.5545 | 22.2406 | 21.0995 | 21.1642 | 22.7338 |
| | | SSIM↑ | **0.7564** | 0.7220 | 0.6494 | **0.7295** | 0.6241 | 0.6355 | 0.5859 | 0.6431 | 0.6029 | 0.5702 | 0.6809 |
| | | LPIPS↓ | **0.2733** | **0.2805** | 0.4383 | 0.3062 | 0.3165 | 0.3053 | 0.3350 | 0.3017 | 0.3190 | 0.3256 | 0.2820 |
| | D2 | CLIPIQA↑ | 0.5518 | 0.5387 | 0.3265 | 0.4700 | 0.4310 | 0.6310 | **0.6337** | 0.5954 | 0.6192 | 0.6337 | **0.7258** |
| | | MUSIQ↑ | 68.7469 | 68.7004 | 37.9649 | 64.6554 | 58.6891 | 70.5298 | **72.4920** | 70.6533 | 70.8595 | 71.8745 | **72.6383** |
| | | MANIQA↑ | 0.5363 | 0.5728 | 0.3294 | 0.4686 | 0.5287 | 0.6085 | 0.6393 | 0.6452 | 0.6230 | **0.6716** | **0.6885** |
| | | PSNR↑ | **21.1313** | 20.4738 | 20.4985 | 20.4978 | 20.7961 | **20.9779** | 20.5045 | 19.8372 | 19.5996 | 20.0026 | 19.5960 |
| | | SSIM↑ | **0.5547** | 0.5496 | 0.4406 | 0.5340 | 0.5378 | 0.5387 | 0.5089 | 0.4797 | 0.5000 | 0.4780 | 0.5131 |
| | | LPIPS↓ | 0.4176 | 0.4092 | 0.5989 | 0.4389 | 0.4046 | 0.4076 | **0.3977** | 0.4117 | 0.4015 | **0.3935** | 0.4863 |
| | D3 | CLIPIQA↑ | 0.4267 | 0.4477 | 0.2316 | 0.3821 | 0.2640 | 0.5858 | **0.6406** | 0.5017 | 0.5497 | 0.6151 | **0.7282** |
| | | MUSIQ↑ | 48.6442 | 51.9957 | 34.4258 | 46.2069 | 30.1708 | 66.7416 | **72.5484** | 61.4580 | 67.8944 | 70.9952 | **72.5719** |
| | | MANIQA↑ | 0.4178 | 0.4757 | 0.1755 | 0.3796 | 0.2684 | 0.5878 | 0.6319 | 0.5549 | 0.5739 | **0.6522** | **0.6867** |
| | | PSNR↑ | **18.9619** | 18.4273 | 18.2784 | 18.4354 | **18.9172** | 18.8488 | 18.1164 | 17.7413 | 17.5175 | 17.7367 | 17.5267 |
| | | SSIM↑ | **0.4303** | 0.4292 | 0.3398 | 0.4078 | 0.3934 | 0.4119 | 0.3751 | 0.3312 | 0.3740 | 0.3381 | 0.3889 |
| | | LPIPS↓ | 0.5371 | 0.5266 | 0.6896 | 0.5482 | 0.6114 | 0.5139 | **0.5051** | 0.5472 | 0.5095 | **0.5057** | 0.5435 |
| *FFHQ-face* | D1 | CLIPIQA↑ | 0.5412 | 0.5567 | 0.2366 | 0.4991 | **0.5962** | **0.6436** | 0.5864 | 0.5303 | 0.5690 | 0.5639 | 0.5026 |
| | | MUSIQ↑ | 74.8917 | 72.2332 | 48.7238 | 71.7499 | 75.5272 | 75.3749 | 75.5116 | 73.1515 | 74.3645 | **75.6899** | 71.8402 |
| | | MANIQA↑ | 0.5724 | 0.5483 | 0.5391 | 0.5282 | 0.5960 | 0.5935 | 0.6003 | **0.6011** | 0.5884 | **0.6339** | 0.5997 |
| | | PSNR↑ | **32.0286** | 30.9152 | 29.7619 | **31.1128** | 28.8965 | 29.8403 | 29.4371 | 30.2578 | 28.2804 | 28.6319 | 30.3368 |
| | | SSIM↑ | 0.8468 | 0.8448 | 0.7031 | **0.8507** | 0.7911 | 0.7866 | 0.7851 | 0.7901 | 0.7467 | 0.7434 | 0.7932 |
| | | LPIPS↓ | **0.2976** | 0.3197 | 0.4548 | 0.3318 | **0.2981** | 0.3426 | 0.3182 | 0.2991 | 0.3309 | 0.3202 | 0.3162 |
| | D2 | CLIPIQA↑ | 0.5198 | 0.4267 | 0.2644 | 0.3401 | 0.5154 | **0.6255** | 0.5425 | 0.5433 | 0.4803 | 0.5716 | **0.5915** |
| | | MUSIQ↑ | 72.4845 | 64.2994 | 26.4976 | 60.7115 | 71.8195 | 73.9450 | 73.2313 | 74.4731 | 70.5953 | **76.1812** | 74.4901 |
| | | MANIQA↑ | 0.5278 | 0.4810 | 0.3630 | 0.4286 | 0.5346 | 0.5887 | 0.5853 | 0.6040 | 0.5640 | **0.6411** | 0.6154 |
| | | PSNR↑ | 28.5038 | 28.2503 | 25.8472 | 28.1374 | **28.5570** | **28.6004** | 27.5172 | 27.7102 | 27.4778 | 27.5357 | 26.9644 |
| | | SSIM↑ | **0.7839** | **0.7887** | 0.5741 | 0.7831 | 0.7699 | 0.7675 | 0.7591 | 0.7403 | 0.7346 | 0.7232 | 0.7329 |
| | | LPIPS↓ | 0.3797 | 0.3699 | 0.5804 | 0.3801 | 0.3602 | 0.3793 | 0.3783 | **0.3593** | 0.3763 | 0.3673 | **0.3559** |
| | D3 | CLIPIQA↑ | 0.3812 | 0.3058 | 0.2367 | 0.2524 | 0.3365 | **0.6157** | 0.5190 | 0.5165 | 0.3928 | 0.5802 | **0.6395** |
| | | MUSIQ↑ | 64.3236 | 53.7270 | 27.0695 | 47.1553 | 52.9726 | 72.4865 | 71.4099 | 73.1891 | 67.3159 | **76.7113** | 76.0612 |
| | | MANIQA↑ | 0.5097 | 0.4468 | 0.1834 | 0.3909 | 0.3783 | 0.5951 | 0.5769 | 0.5858 | 0.5438 | **0.6467** | 0.6302 |
| | | PSNR↑ | **25.9960** | 25.7447 | 23.0445 | 25.3520 | 24.8549 | 25.8737 | 24.8490 | 24.7313 | 24.4744 | 23.9652 | 24.2035 |
| | | SSIM↑ | 0.6992 | **0.7144** | 0.5433 | **0.6966** | 0.6263 | 0.6710 | 0.6684 | 0.6210 | 0.6270 | 0.5930 | 0.6330 |
| | | LPIPS↓ | **0.4258** | 0.4495 | 0.6197 | 0.4641 | 0.5090 | 0.4584 | **0.4334** | 0.4351 | 0.4404 | 0.4349 | 0.4354 |

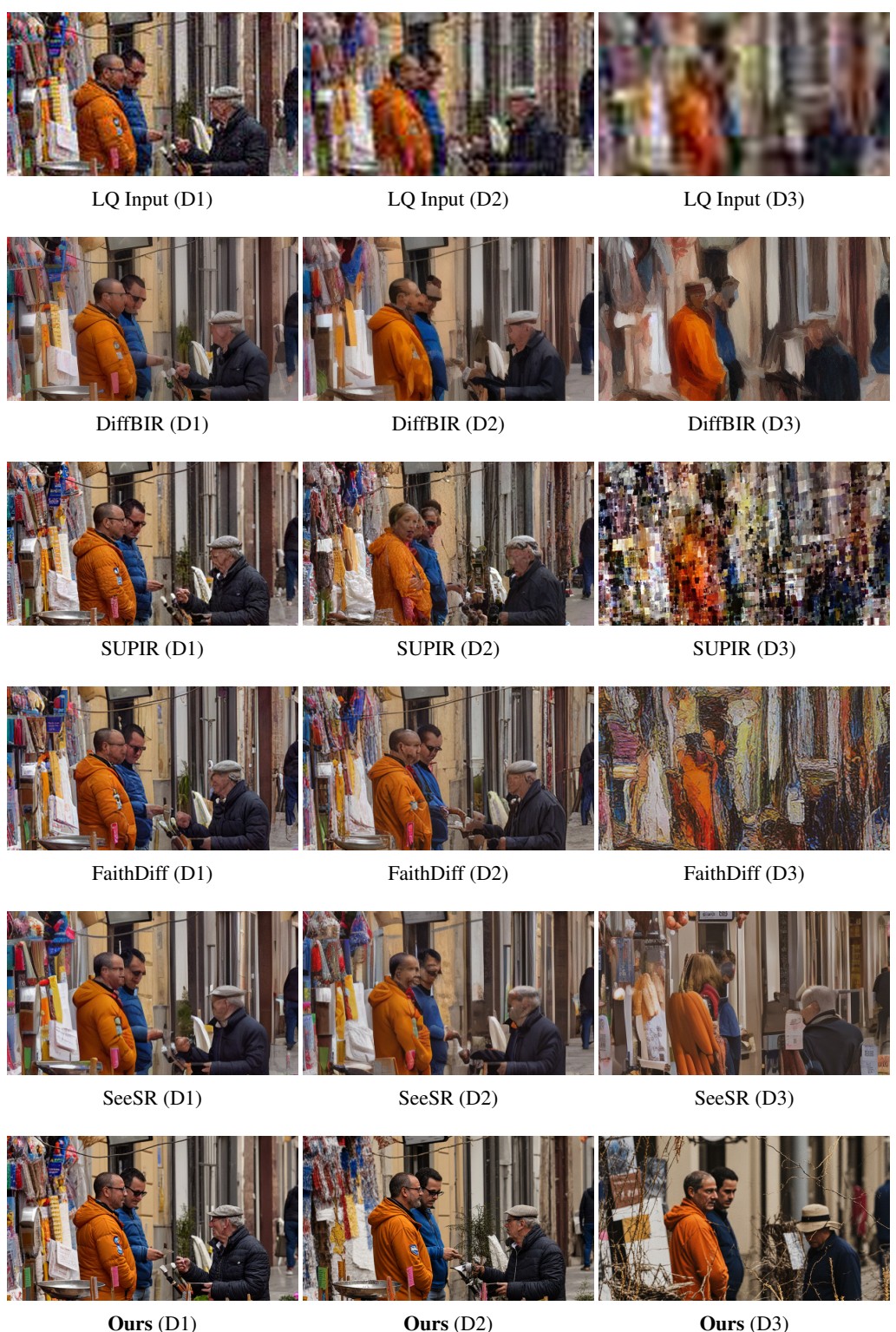

Figure 7: **Qualitative comparisons of different degradation level on LSDIR-Val.** Among diffusion-based approaches, our method achieves the strongest structural fidelity, accurately preserving geometric and semantic details under severe degradations. It reconstructs high-frequency textures and suppresses noise more effectively than competing methods, maintaining both global structural alignment and localized perceptual coherence.

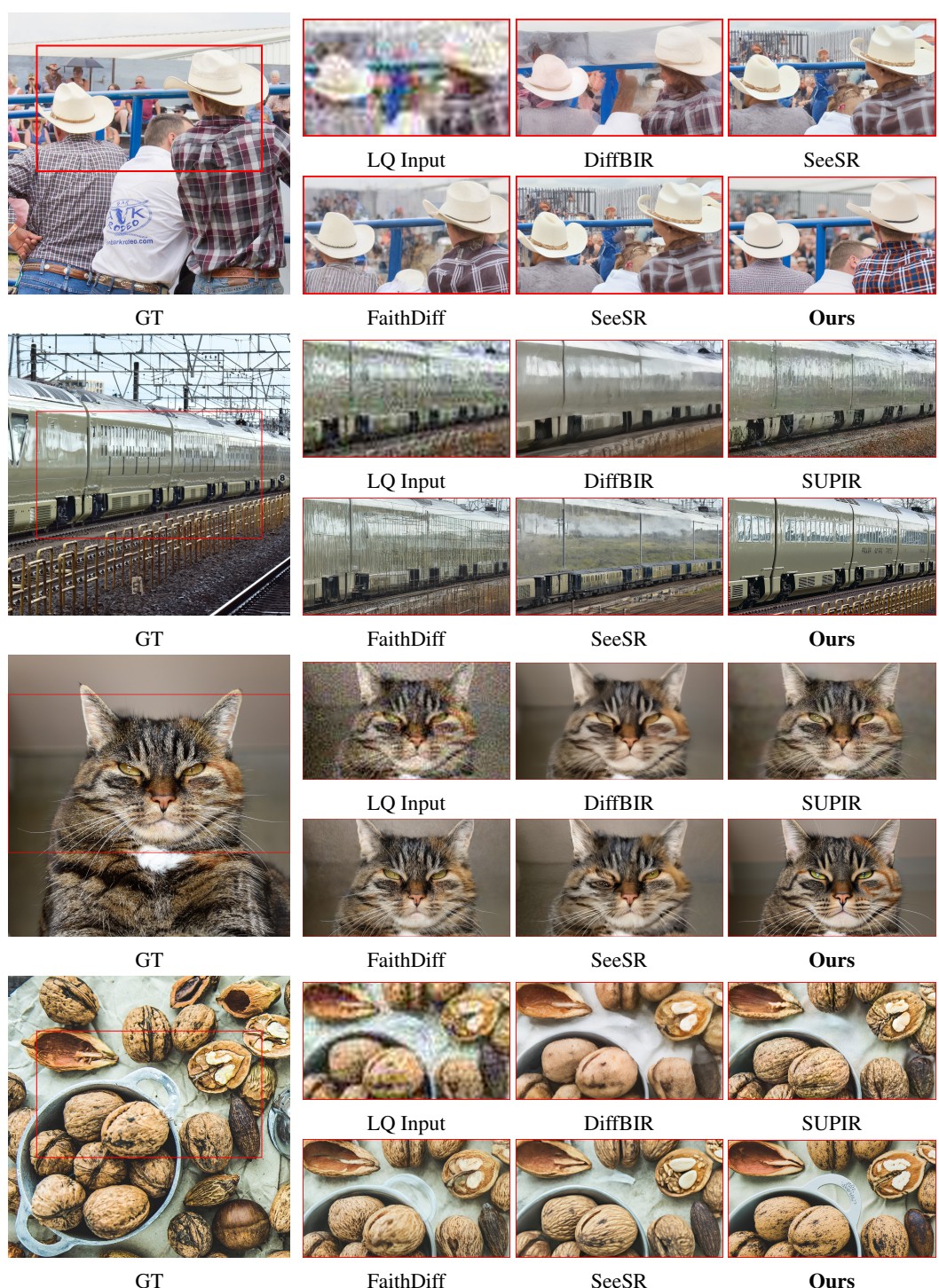

Figure 8: **Qualitative comparisons on DIV2K-Val (D3).** Among diffusion-based approaches, our method achieves the strongest structural fidelity, accurately preserving geometric and semantic details under severe degradations. It reconstructs high-frequency textures and suppresses noise more effectively than competing methods, maintaining both global structural alignment and localized perceptual coherence.

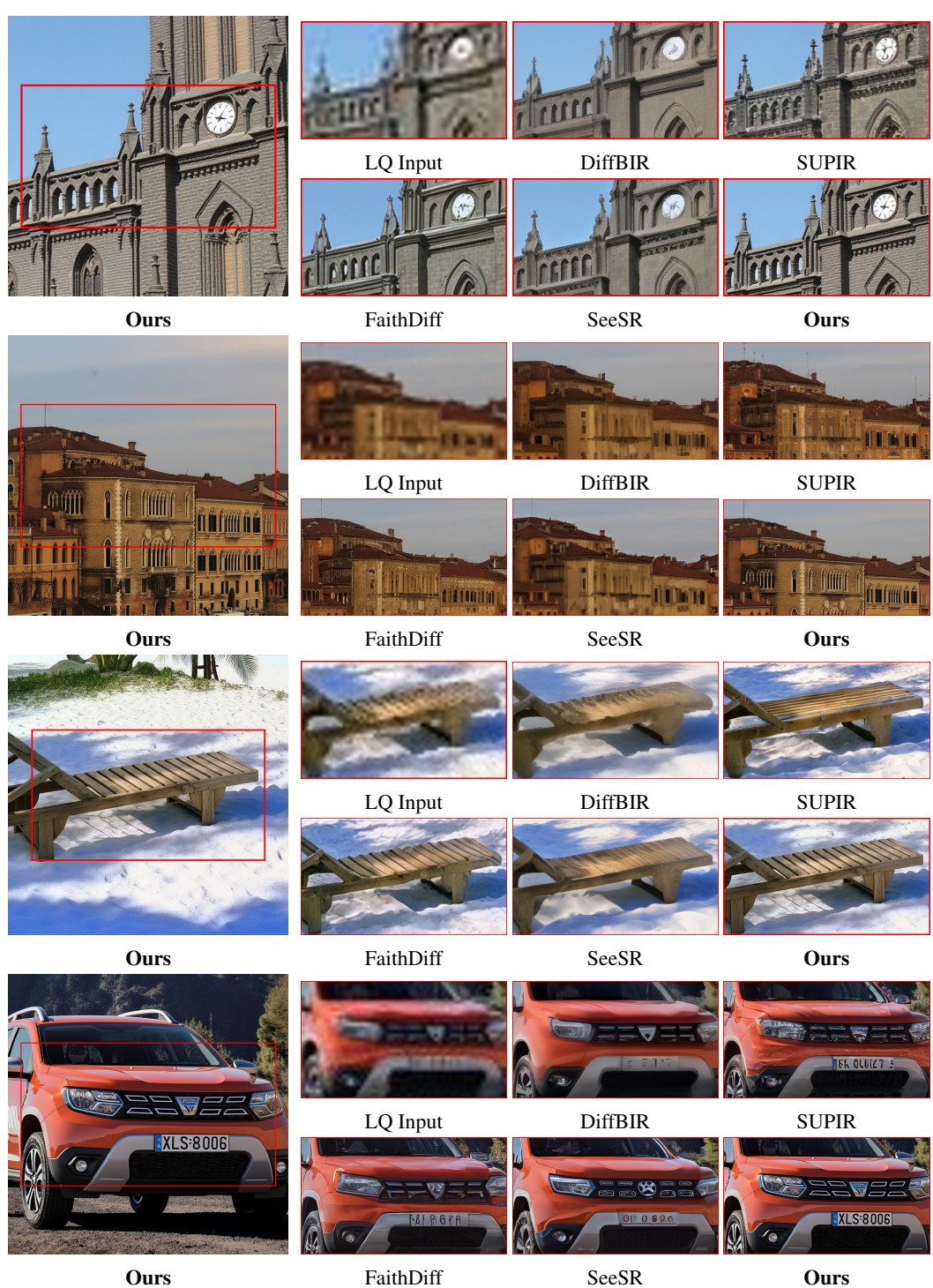

Figure 9: **Qualitative comparisons on RealPhoto60 (2×).** Left: our full-resolution result with crop locations marked (red boxes). Right: grayscale input and outputs from Real-ESRGAN, DiffBIR, FaithDiff, SeeSR, and ours on the corresponding crops.

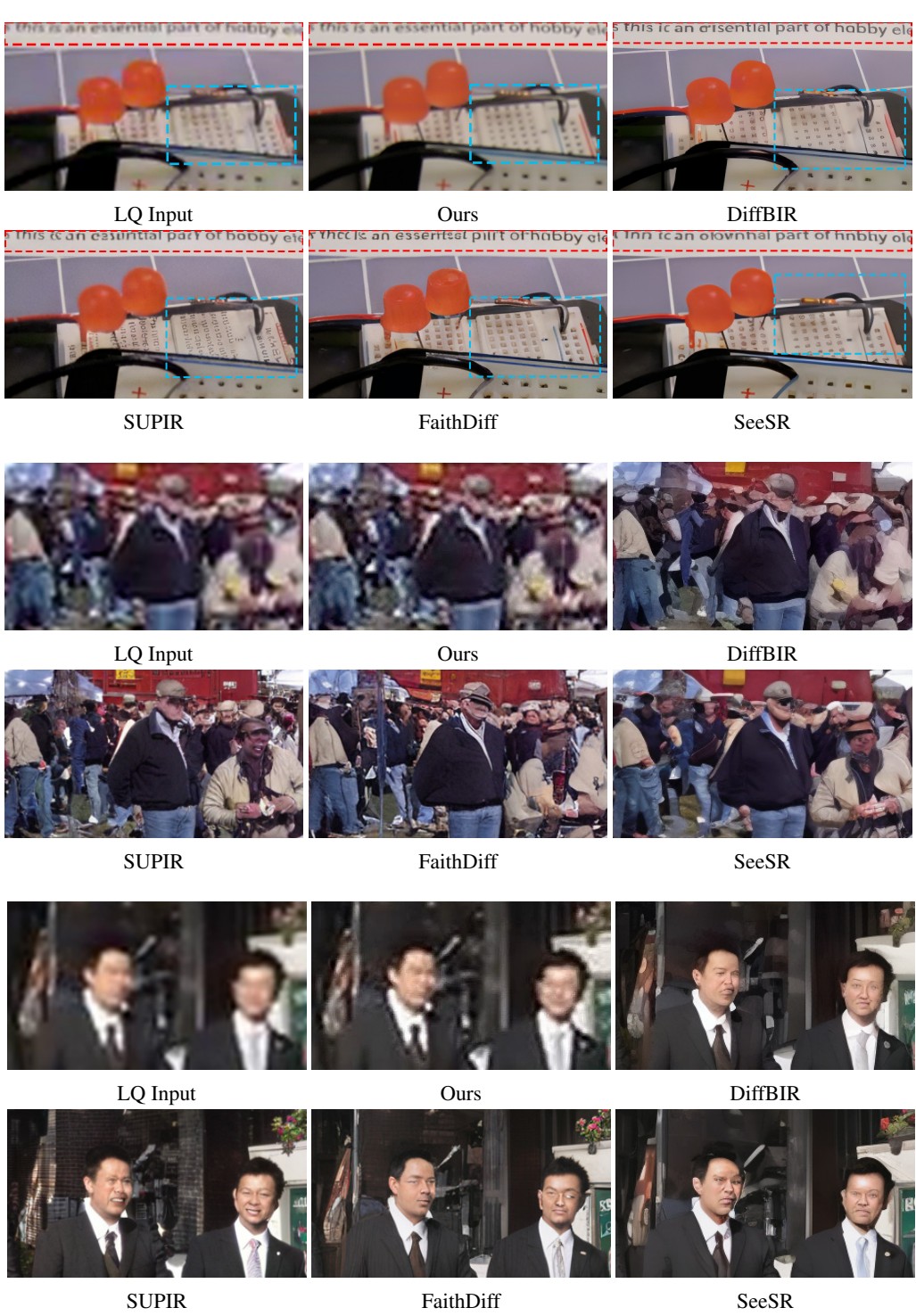

Figure 10: **Failure cases on RealLQ250.** In long-tail cases with degradations outside the scope of our degradation model (downsampling, blur, Gaussian noise, JPEG compression), our framework adopts a reliability-oriented strategy, producing results that remain closer to the LQ input. Competing methods may yield visually sharper outputs but often introduce misleading artifacts under high uncertainty. This conservative behavior ensures restoration that is both high-fidelity and reliable, which is particularly important in applications like restoration of historical artworks.

Table 9: **Quantitative experiments in "conservative" strategy on RealHQ250.** We perturbed RealLQ250 inputs by applying a Gaussian blur with a standard deviation equal to 1% of the image width. If our method were merely "overfitting" or "anchoring" to the low-quality input, this additional degradation would be expected to further reduce performance. Instead, we observe a clear improvement across all no-reference metrics.

| RealHQ250 | CLIPIQA↑ | MUSIQ↑ | MANIQA↑ |
|-----------|----------|--------|---------|
| Origin | 0.4792 | 54.3427 | 0.5397 |
| Blurry | **0.5041** | **60.5381** | **0.5605** |

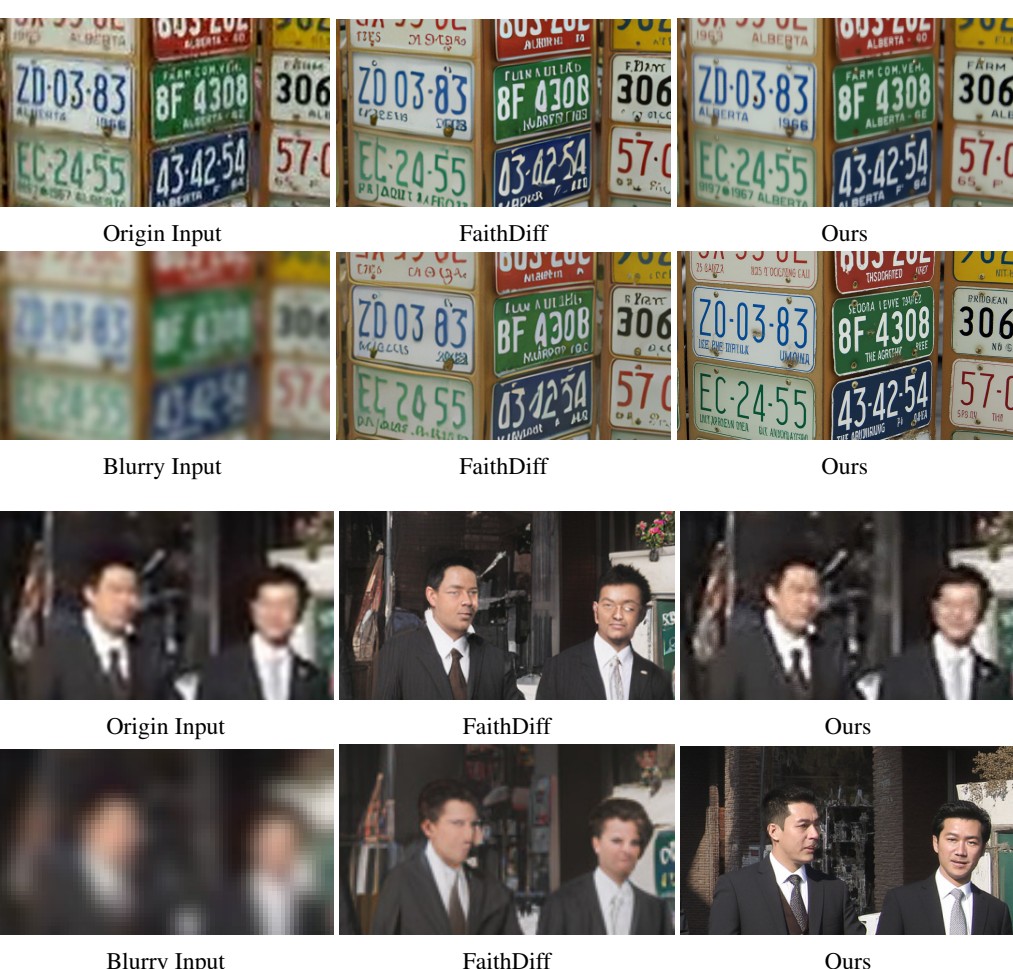

Figure 11: **Qualitative experiments in "conservative" strategy on RealHQ250.** We perturbed RealLQ250 inputs by applying a Gaussian blur with a standard deviation equal to 1% of the image width. If our method were merely "overfitting" or "anchoring" to the low-quality input, this additional degradation would be expected to further reduce performance, as is typical for diffusion-based restoration methods. Instead, we observe a clear improvement in perceptual quality.

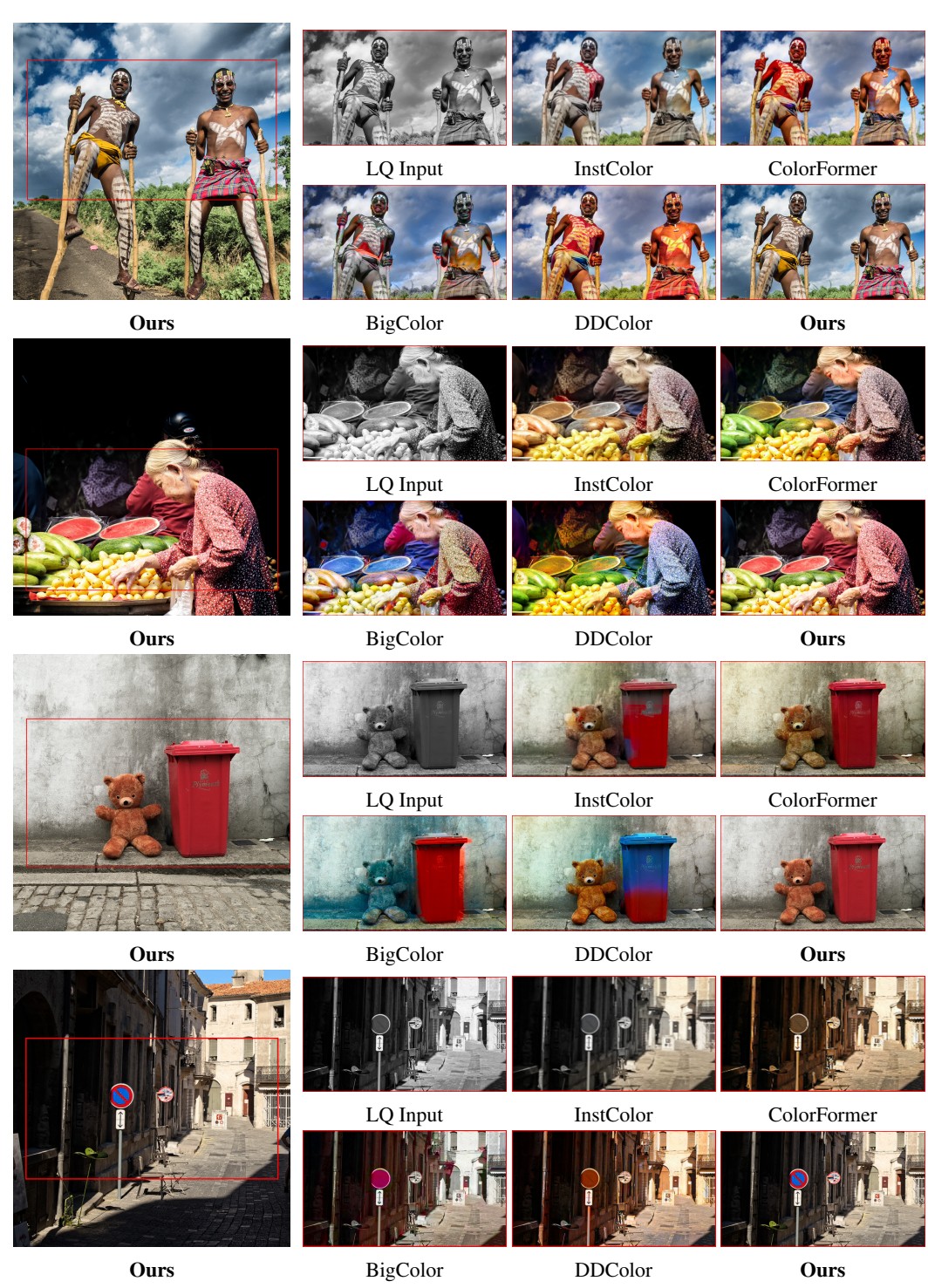

Figure 12: **Qualitative comparisons on LSDIR (colorization).** Left: our full-resolution result with crop locations marked (red boxes). Right: grayscale input and outputs from InstColor, ColorFormer, BigColor, DDColor, and ours on the corresponding crops.

