# OpenReview forum: "In-Token Learning for High-Fidelity Image Restoration via Diffusion Transformers"
_ICLR.cc/2026/Conference — Submitted to ICLR 2026_

### Official Review · Reviewer_zDzX · 2025-10-27

**Soundness:** 3
**Presentation:** 2
**Contribution:** 1
**Rating:** 2
**Confidence:** 5

**Summary:**

This paper introduces In-Token Learning, a novel token-aligned framework for robust image restoration using diffusion models. The approach redefines restoration as learning a conditional velocity field via rectified flow matching within a Multimodal Diffusion Transformer (MMDiT), enabling direct and accurate transport from noise to clean images. To stabilize conditioning, the authors propose Direct Low-Quality Guidance (DLG), which efficiently integrates degraded-image and prompt embeddings into the model’s text-conditioning pathway, eliminating the need for external prompts or complex architectures.

**Strengths:**

1. The proposed method is simple and effective.
2. The paper is easy to understand and follow.
3. The experiments are comprehensive, which contain SR, deoising, and colorization.

**Weaknesses:**

1. The proposed method is overly simplistic, as it merely concatenates the low-quality image to the input of FLUX and applies LoRA for fine-tuning, without any technical innovation.
2. The results of the ablation experiments show excessively large performance differences among the various settings, which makes me question the authenticity of these ablation data. I do not believe that these ablated components could have such a significant impact on PSNR, SSIM, and CLIPIQA metrics. I strongly recommend that the authors provide reproducible code and models to enhance the credibility of the paper.
3. There are obvious errors in the citation format throughout the paper.
4. The paper does not provide efficiency comparisons with other methods, such as parameter count and inference speed.

**Questions:**

Refer to Weakness

---

> ### Author Response · Authors · 2025-11-16
> **Response to Reviewer zDzX**
>
> We appreciate the reviewer’s careful reading and address the points below.
>
> ## ▌Response to w1 — Contribution of In-Token Learning
>
> ITL is not a concatenation heuristic. It is structurally required because, after redefining the RFM endpoint as
> $$
> z_0 = E(D^{-1}\_\phi (I\_{lq})),
> $$
> our formulation has **no diffusion prior** to rely on. The model must therefore learn a new clean-image manifold induced solely by the inverse-degradation distribution, fundamentally different from standard diffusion finetuning.
>
> Under this endpoint, the degraded evidence $E(I\_\mathrm{lq})$ must influence the velocity field **without ever entering the flow trajectory**. Among all conditioning mechanisms, In-Token Learning (ITL) is the only one that satisfies the structural constraints imposed by rectified-flow restoration:
>
> 1. injecting degraded information token-wise to guide the inverse mapping
> 2. **without contaminating the latent trajectory $z_t$** with degraded statistics
> 3. preserving **fixed-length attention**, enabling 4K/8K/12K scalability
> 4. remaining compatible with LoRA-only training in the absence of a diffusion prior
>
> Alternative designs—sequence concatenation, ControlNet-style branches, or denoising-based conditioning， violate one or more of these constraints by either mixing degraded features into $z_t$, breaking fixed-length attention, or destabilizing the inverse-degradation mapping.
>
> Thus, ITL is not an architectural convenience or a renamed concatenation scheme. It is the **minimal conditioning structure** that makes our restoration-specific rectified-flow endpoint feasible. The novelty lies in
> 1.  the **formulation (new endpoint and target distribution)** and
> 2.  the **structural role ITL plays within it**,
>
> not in the operation of concatenation itself.
>
> ## ▌Response to w2 — Ablations and reproducibility
>
> The large ablation gaps arise because LoRA is learning a new clean-image prior from scratch, not adjusting a diffusion prior. This naturally makes conditioning design critical. Removing ITL or DLG breaks either the structural or semantic prior, causing the LoRA-updated velocity field to drift or collapse.
>
> The results are fully reproducible using the provided scripts and configurations. **All inference code, model checkpoints, and raw ablation outputs** are included in the anonymous repository.
>
> ## ▌Response to w3 — Citation formatting
>
> We thank the reviewer for pointing this out.
> All references have been carefully corrected in the revised manuscript to follow ICLR style guidelines.
>
> ## ▌Response to w4 — Efficiency comparisons
>
> We have added an explicit efficiency comparison in the appendix, including both **trainable parameter count** and **1536×1536 inference speed** for major recent methods. All models are evaluated using **28 inference steps**, and each method is run under its recommended GPU configuration for fair comparison. Notably, **FaithDiff, DreamClear, and SUPIR all require two RTX 5880 Ada GPUs** for inference at this resolution, whereas **our method runs on a single RTX 5880 Ada**, reflecting substantially lower hardware requirements.
>
> | Method     | Trainable Params | Inference Speed (1536²) | GPU Requirement |
> | ---------- | ---------------- | ----------------------- | --------------- |
> | SeeSR      | 2.3B             | 34 s                    | **1× 5880 Ada** |
> | SUPIR      | 1.3B             | **26 s**                | 2× 5880 Ada     |
> | DreamClear | 2.4B            | 107 s                   | 2× 5880 Ada     |
> | FaithDiff  | 2.4B             | 35 s                    | 2× 5880 Ada     |
> | **Ours**   | **1B**           | 75 s                    | **1× 5880 Ada** |
>
> Achieving strong performance across multiple tasks with **only 1B trainable parameters**, **running on a single GPU**, and **without relying on diffusion priors**, further indicates that the method is not incremental but reflects a fundamentally different restoration formulation.
>
> While certain diffusion-based models achieve faster 1536² sampling through more aggressive denoising schedules, they do so with **larger models and higher GPU requirements**. In contrast, our formulation prioritizes **stability and native scalability to 4K–12K**, which is the primary design goal for this work.

---

> > ### Author Response · Authors · 2025-11-27
> >
> > Dear Reviewer,
> >
> > Thank you again for your review. I hope our rebuttal has addressed the points you raised.
> > If any part would benefit from additional clarification, we would be very happy to provide it.
> >
> > Thank you for your time.

---

### Official Review · Reviewer_t7Y5 · 2025-10-27

**Soundness:** 2
**Presentation:** 3
**Contribution:** 2
**Rating:** 2
**Confidence:** 4

**Summary:**

This paper proposes a strategy of learning the conditional velocity field via rectified flow matching, directly generating clean images from noisy with low-quality and text prompts as conditions. It achieves good image restoration quality and has a certain generalization effect for larger image resolutions.

**Strengths:**

1.The writing is clear and easy to understand, facilitating readers' comprehension.

2.The proposed method has consistently stable restoration results for the restoration tasks at different resolutions.

**Weaknesses:**

1.The evaluation metrics of the paper are incomplete.  LPIPS remains the main metric for assessing the quality of image restoration. However, it is not reported in the main text of the paper, and the LPIPS, PSNR, and SSIM under all settings are not reported in the supplementary materials either.
2. The additional low-quality and text prompts used as conditions seem more like an engineering trick to enhance the ability of condition control..  Moreover, the specific role these text prompts play in the text is not clear; it might merely be to match the pattern of the pre-trained model.
3. There is an issue of unfairness in the experimental setup. Why should the test metrics of different settings of super-resolution be averaged on the basis of no-reference metrics, but only full-reference metrics of D1 be reported? This will cause misunderstandings for the readers.Furthermore, the experimental setup is also different from other methods. Other methods were not trained on different degraded images. Such a comparison is difficult to determine whether it is the effect of the network design or the training settings.

**Questions:**

Please refer to the weaknesses.

---

> ### Author Response · Authors · 2025-11-16
> **Response to Reviewer t7Y5**
>
> We thank the reviewer for the constructive feedback.
>
> ## ▌Response to w1 — Completeness of evaluation metrics
>
> As suggested, we have added LPIPS results to the main paper for all super-resolution settings.
> We additionally provide full PSNR, SSIM, and LPIPS results for all degradation levels (D1/D2/D3) in the supplementary materials as requested.
> This ensures complete and transparent evaluation.
>
> ## ▌Response to w2 — Necessity of DLG
>
> Because our formulation discards diffusion priors by redefining the RFM endpoint, LoRA must learn the clean manifold without denoising refinement. This demonstrates that **LoRA** training here is not incremental, it learns a new clean-image prior induced by our inverse-degradation endpoint, rather than adjusting any pretrained diffusion prior.
>
> DLG is introduced to provide semantic priors for the inverse-degradation target. The conditional RFM objective is:
> $$
> \mathcal{L}\_{\mathrm{RFM}}
> = \mathbb{E}\left[\|v\_{\theta}(z_t, t, e_f, [x_t;\,E(I\_{\mathrm{lq}})]) - (\epsilon - E(\mathcal{D}^{-1}\_{\phi}(I\_{\mathrm{lq}})))\|_2^2\right].
> $$
> DLG provides:
>
> - a **task-level prior** (system prompt) describing the high-level semantics of the inverse process $\mathcal{D}^{-1}\_{\phi}$, and
> - an **instance-level prior** (image embeddings) describing degradations in $I_{\mathrm{lq}}$.
>
> ## ▌Response to w3 — On fairness of the experimental setup
>
> Our training and evaluation protocol directly follows **FaithDiff (CVPR 2025)**, which also trains on multiple degradations and evaluates across D1–D3.
>
> This alignment ensures strict comparability with recent baselines and avoids introducing evaluation bias.
>
> To eliminate any ambiguity, we now report full-reference metrics (PSNR/SSIM/LPIPS) for all degradation levels in the supplementary materials.

---

> > ### Author Response · Authors · 2025-11-27
> >
> > Dear Reviewer,
> >
> > If there is any part of our response where additional detail or explanation would be
> > useful, we would be happy to provide supplementary information.
> >
> > Thank you very much for your time and thoughtful feedback.

---

### Official Review · Reviewer_wYA2 · 2025-10-29

**Soundness:** 4
**Presentation:** 3
**Contribution:** 4
**Rating:** 8
**Confidence:** 4

**Summary:**

In this paper, the authors propose "In-Token Learning", a unified paradigm for high-fidelity image restoration, with five core contributions:  first, it innovates the restoration paradigm by abandoning the traditional "iterative denoising of degraded images" approach, learning a conditional velocity field via Rectified Flow Matching (RFM), and combining in-token alignment with Direct Low-Quality Guidance (DLG) to achieve "direct mapping from pure noise to clean images", thus avoiding the propagation of degraded artifacts;  second, it designs the lightweight guidance mechanism DLG, which injects the fused embedding of degraded images and task prompts into the model's native text-conditioning pathway without relying on external Vision-Language Models (VLMs) or ControlNet-style side branches, providing task-aware guidance at minimal cost;  third, it narrows the perception-distortion gap by simultaneously improving full-reference metrics (PSNR, SSIM) and no-reference perceptual metrics (CLIPIQA, MUSIQ) across multiple benchmark datasets (DIV2K, LSDIR, etc.) and different degradation scenarios, alleviating the problem of "mismatch between perceptual effects and objective distortion";  fourth, it supports ultra-high resolutions: leveraging the fixed-length attention mechanism of in-token alignment, it natively enables direct inference at QHD (2560×1440) resolution, achieves 4K/8K/12K resolution restoration through tile-consistent expansion, and verifies this capability with the 12K restoration of the classical scroll painting Along the River During the Qingming Festival; fifth, it exhibits strong task generalization: the same backbone network and training pipeline can seamlessly extend from super-resolution tasks to image colorization tasks without re-designing the model, demonstrating excellent cross-task transfer performance.

**Strengths:**

The proposed "intra-token alignment + RFM" restoration paradigm differs significantly from existing "iterative denoising" or "sequence-level conditional fusion" methods, representing a novel technical route.
The DLG mechanism cleverly leverages the model’s native text pathway, avoiding dependencies on external models and additional branch overhead. It balances "conditional constraint strength" and "computational efficiency" with an innovative design.
The experimental design is rigorous, covering "synthetic + real-world," "low-resolution + ultra-high-resolution," and "single-task + cross-task" scenarios.  Ablation experiments (on DLG components, generation modes, and token alignment) are comprehensive, ensuring highly credible results.
The author demonstrate the first 12K restoration of the classical scroll painting Along the River During the Qingming Festival. Ultra-high-resolution scalability provides an efficient solution for 12K restoration, with significant application value in fields such as historical artifact restoration.

**Weaknesses:**

The paper claims that in-token fusion and DLG “bridge the perception-distortion gap” and “stabilize conditioning,” yet no clear mathematical or causal analysis is provided to substantiate these effects.
The “Direct Low-Quality Guidance” module (DLG) is described as fusing embeddings of the degraded image and system prompt — yet it remains unclear what the “system prompt” represents, or how it differs across tasks (SR, denoising, colorization).

**Questions:**

In Figure 3, the person in the LR input appears to have no two front teeth, yet the restored image incorrectly adds two front teeth. Does this indicate a lack of fidelity?

---

> ### Author Response · Authors · 2025-11-16
> **Response to Reviewer wYA2**
>
> We thank the reviewer for the constructive evaluation. Below we clarify the causal role of DLG and the perception–distortion behavior, highlighting that our formulation does not rely on diffusion priors.
> ## ▌Response to w1 — Role of DLG and perception–distortion
>
> DLG is essential because our method has **no diffusion denoising prior** to stabilize the inverse mapping.
> DLG is introduced to provide semantic priors for the inverse-degradation target. The conditional RFM objective is:
> $$
> \mathcal{L}\_{\mathrm{RFM}}
> = \mathbb{E}\left[\|v\_{\theta}(z_t, t, e_f, [x_t;\,E(I\_{\mathrm{lq}})]) - (\epsilon - E(\mathcal{D}^{-1}\_{\phi}(I\_{\mathrm{lq}})))\|_2^2\right].
> $$
> DLG provides:
>
> - a **task-level prior** (system prompt) describing the high-level semantics of the inverse process $\mathcal{D}^{-1}_{\phi}$, and
> - an **instance-level prior** (image embeddings) describing degradations in $I_{\mathrm{lq}}$.
>
> In practice, the system prompt simply specifies the semantic intent of the inverse process (e.g., “produce a clean high-quality image” for SR&Denoise and “colorize this grayscale input” for colorization). These prompts are fixed and not engineered.
>
> This combination enables the model to recover sharp, coherent structure while avoiding hallucination, explaining the simultaneous improvement in PSNR/SSIM and perceptual scores (CLIPIQA/MUSIQ) and thereby narrowing the perception–distortion gap.
>
> ## ▌Response to Q1 — Fidelity in Figure 3
>
> The LR mouth region in Figure 3 is highly ambiguous; although the downsampled image does not clearly show two front teeth, it still contains faint high-frequency cues. When the degradation only partially matches $\mathcal{D}_{\phi}$, the inverse mapping is not unique, and several plausible clean reconstructions may exist.
>
> - If the degradation is well described by $\mathcal{D}\_{\phi}$, the inverse mapping becomes unambiguous.
> - If the degradation lies outside $\mathcal{D}\_{\phi}$, the model behaves conservatively (see page 21 of revised PDF for more experiments).
> - In intermediate cases, the model selects the **most plausible clean structure consistent with the available evidence**, not arbitrary hallucinations.
>
> This multi-solution behavior is not unique to our method; under such severe ambiguity, **all restoration models tend to recover a structurally plausible pair of front teeth**. The difference is that our formulation produces a reconstruction with **higher structural coherence and better global visual consistency**, as confirmed by additional comparisons in the revised appendix. This reflects the inherent uncertainty of the inverse problem in this regime.

---

> > ### Author Response · Authors · 2025-11-27
> >
> > Dear Reviewer,
> >
> > Thank you again for your thoughtful and detailed review. Your insights captured the
> > core motivation and contributions of our work very accurately, and we greatly
> > appreciate the time you invested in understanding the formulation and experiments.
> >
> > If any part of our rebuttal would benefit from additional clarification or further
> > evidence, we would be more than happy to provide it.
> >
> > Thank you once again for your support and for helping improve the paper.

---

### Official Review · Reviewer_uxXd · 2025-11-01

**Soundness:** 2
**Presentation:** 3
**Contribution:** 2
**Rating:** 2
**Confidence:** 3

**Summary:**

This paper proposes an image restoration framework based on Diffusion Transformer (DiT) and Rectified Flow Matching (RFM). The authors define their approach as learning a conditional velocity field that transports pure noise directly to a clean image. The framework is a combination of three existing components: 1) RFM as the generative paradigm; 2) "In-Token Learning" (ITL), which is channel-wise concatenation of conditions; and 3) "Direct Low-Quality Guidance" (DLG), a multi-modal cross-attention injection mechanism.

The authors claim SOTA performance on multiple benchmarks, a reduction in the perception-distortion gap, and scalability to 12K resolution inference.

**Strengths:**

1. Good Empirical Results: The proposed system achieves (parts of) SOTA or highly competitive performance on several synthetic image restoration benchmarks.
2. Promising Scalability: The method is successfully demonstrated on ultra-high-resolution 12K imagery, which is a notable engineering achievement. The underlying channel-wise (ITL) approach is indeed more computationally scalable than sequence-wise concat.

**Weaknesses:**

1.  Overstated Contribution:
    * The paper's core flaw is over-packaging. The central contribution "In-Token Learning" is a standard channel-wise concatenation (where is "Learning"?), and "DLG" is a standard multi-modal attention injection. The paper re-brands these existing techniques with new nomenclature, supported by a trivial complexity analysis (Sec. 3.8), which obscures a lack of methodological novelty.
    * And also, there is a disconnect between the paper's central argument and its methods. The claim of a new paradigm ("transporting pure noise directly to a clean image") is merely a description of the underlying RFM. The paper fails to articulate the necessary link between this high-level concept and the specific combination of the ITL and DLG mechanisms (at least from my view). Please clarify this.
2.  Incremental System-Building Work: This work is like an incremental systems-building project, combining three existing components (RFM, channel-concat, cross-attention) well. However, it lacks a strong **justification for the synergy** of this specific combination (i.e., "Why these three?") and offers no new fundamental principles for representation learning, which is the focus of ICLR.
3.  Not Good Real-World Generalization: Despite strong synthetic results, the method significantly **underperforms** all established baselines on **all metrics** on the real-world RealLQ250 dataset. The defense of this as a "conservative" strategy (Sec. 4.2) is unconvincing and more likely masks overfitting to the synthetic degradation model $\mathcal{D}_{\phi}$.

**Questions:**

1. Contradictory Ablation: In the Table 4 ablation, you show that "anchoring to the flawed LQ input" (e.g., "Denoise 0.9") degrades performance. Yet, the core ITL method strongly "anchors" to the flawed LQ latent $y$ at *every* step via $h_t = [x_t; y]$. Please clarify the fundamental difference between these two settings and explain why ITL is not negatively affected.

2. Justification for 'Conservatism': The method underperforms on the RealLQ250 dataset, which you attribute to a "reliability-oriented strategy" that "conservatively" avoids hallucination. This justification appears to be a post-hoc rationalization for poor generalization, likely due to overfitting on your synthetic degradation model $\mathcal{D}_{\phi}$. What concrete evidence can you provide to support that this is a beneficial 'conservative' behavior rather than simply a model failure on out-of-distribution real-world data?

3. Necessity of DLG's Prompt Content: How critical is the *semantic content* of the "system prompt" in DLG? What is the performance if $e_t$ is replaced with a null-text or random embedding? The current ablation (Table 3) only removes $e_t$ entirely, which fails to decouple its presence from its semantic meaning. I have doubts about the contribution/effect of the specific text content itself (as shown in Figure 1, prompts like "Produce clean, sharp, noise-free images..." seem empty and lack information). Please do a Placebo Experiment ([empty text + LQ] vs [text + LQ]) within the same Text Encoder structure, to see the benefit come from semantic guidance or from an **unexplored architectural bias**.

---

> ### Author Response · Authors · 2025-11-16
> **Response to Reviewer uxXd**
>
> We thank the reviewer for the detailed analysis. Below we clarify the novelty of our formulation, the necessity of ITL and DLG, and why our model’s behavior cannot be attributed to diffusion priors.
>
> ## ▌ Response to W1 — On methodological novelty and the role of ITL and DLG
>
> Our work does **not** inherit diffusion priors. We redefine the RFM endpoint as
> $$
> z_0 = E(\mathcal{D}^{-1}\_{\phi}(I\_{\mathrm{lq}})),
> $$
> the model no longer transports noise toward the clean manifold learned during diffusion pretraining.
>
> Instead, LoRA must learn a **new** clean-image prior from the contracted inverse-degradation distribution.
> ITL supplies structural evidence **without injecting degraded statistics into $z_t$**, while DLG provides semantic guidance needed for learning $\mathcal{D}^{-1}_{\phi}$ from scratch.
> Removing either leads to unstable or severely degraded velocity-field learning.
>
> Thus, the novelty lies not in architectural additions but in a **conditional RFM formulation that requires new target distributions, new conditioning structures, and new learned priors**.
>
> ## ▌Response to W2 — Why RFM, ITL, and DLG form a necessary combination
>
> > The conditional RFM objective can be written as:
>
> $$
> \mathcal{L}\_{\text{RFM}} = \mathbb{E}\_{z\_0,\epsilon,t}\Big[\\|v\_\theta(z\_t,t,e\_f, [x\_t; E(I\_{lq})]) - (\epsilon-E(\mathcal{D}^{-1}_\phi(I\_\text{lq})))\|_2^2\\Big].
> $$
>
> Because our formulation discards the diffusion denoising prior, these components are not interchangeable but **structurally required**:
>
> - **RFM** defines a stable deterministic transport from pure noise $\epsilon$ to the restoration-specific endpoint $z\_0 = E(\mathcal{D}^{-1}_{\phi}(I\_{\mathrm{lq}}))$ , which replaces the clean-image prior normally inherited from diffusion.
>
> - **ITL** injects structural evidence through $h_t = [x_t; E(I\_{\mathrm{lq}})]$, while keeping degraded statistics out of the trajectory and preserving fixed-length attention.
>
> - **DLG** supplies the semantic priors needed to learn $\mathcal{D}^{-1}_{\phi}$ from scratch: the system prompt provides a **task-level semantic prior**, and the image embedding provides an **instance-level semantic prior** for $I\_{\mathrm{lq}}$.
>
> ## ▌Response to W3 — On “conservative” behavior vs. overfitting
>
> If the model were overfitting to $\mathcal{D}\_{\phi}$, additional blur should worsen results, as is typical for diffusion-based restorers.
>
> However, adding a mild 1% Gaussian blur, which slightly moves $I\_{\mathrm{lq}}$ back into the support of $\mathcal{D}\_{\phi}$, improves all no-reference metrics (page 21 of the revised PDF):
> $$
> \mathcal{D}^{-1}\_{\phi}(I\_{\mathrm{lq}}) \approx I\_{\mathrm{lq}} \quad \Rightarrow \quad \text{conservative output}.
> $$
> The **improvement after adding blur** is therefore incompatible with the overfitting hypothesis and matches the predicted behavior of our conditional RFM formulation.
>
> ## ▌Response to Q1 — Why ITL does not anchor the model to $y$
>
> The denoise-$\alpha$ variants modify the trajectory by mixing $y$ into $z\_t$, causing degraded statistics to propagate.
> ITL does not alter $z\_t$; it only modulates the velocity field via:
> $$
> h\_t = [x\_t ; y].
> $$
> Thus, ITL affects conditioning but never anchors the generative state, which explains why ITL improves performance while denoise-$\alpha$ degrades it.
>
> ## ▌Response to Q2 — Additional evidence of principled conservatism
>
> We perturbed RealLQ250 inputs by **applying a Gaussian blur** with a standard deviation equal to 1% of the image width. If our method were merely “overfitting” or “anchoring” to the low-quality input, this additional degradation would be expected to further reduce performance, as is typical for diffusion-based restoration methods.
>
> Instead, we observe a clear **improvement in perceptual quality**. Thus, the RealLQ250 results reflect the expected conservative behavior of our conditional-RFM design under degradation mismatch, not a failure of generalization.
>
> We include these results in the revised appendix:
>
> | Input Type           | CLIPIQA    | MUSIQ       | MANIQA     |
> | -------------------- | ---------- | ----------- | ---------- |
> | RealHQ250 (original) | 0.4792     | 54.3427     | 0.5397     |
> | RealHQ250 (+1% blur) | **0.5041** | **60.5381** | **0.5605** |
>
>
> ## ▌Response to Q3 — Placebo experiments for DLG
>
> As suggested by the reviewer, we evaluated two placebo variants:
>
> - **Empty-text embedding:** nearly identical results due to training-time dropout with empty text.
> - **Random embedding:** severe collapse in results because random semantics disrupt the inverse-degradation prior.
>
> | DLG variant                     | PSNR        | SSIM       | CLIPIQA    |
> | ------------------------------- | ----------- | ---------- | ---------- |
> | **Full DLG**                    | 21.9795     | **0.5698** | **0.6293** |
> | empty text (placebo)            | **21.9809** | 0.5698     | 0.6273     |
> | random text embedding (placebo) | 7.9733      | 0.0438     | 0.3481   |

---

> > ### Author Response · Authors · 2025-11-27
> >
> > Dear Reviewer,
> >
> > Thank you again for the detailed comments. If any part of our response would benefit
> > from further clarification or additional evidence, we would be very happy to provide it.
> > Please let us know if there is anything else you would like us to address.
> >
> > We sincerely appreciate your time and help in improving the paper.

---

### Author Response · Authors · 2025-11-16
**General Response**

We thank all reviewers for their thoughtful feedback. A recurring assumption in several comments is that our method inherits strong generative priors from diffusion models and that LoRA fine-tuning merely adjusts those priors. This is not the case. Our work does **not** rely on any diffusion denoising prior. Instead, our core contribution is a **restoration-specific redefinition of Rectified Flow Matching (RFM)** that fundamentally changes the target distribution and the geometry of the generative trajectory.

## ▌1. Redefining the RFM endpoint removes diffusion priors

Standard RFM (and diffusion) transport noise toward a clean-image manifold learned during pretraining.
We instead redefine the endpoint as the inverse-degradation target:

$$
z_0 = E(D^{-1}_\phi (I\_{lq}) )
$$

which contracts the target distribution and grounds it in the degradation model rather than the diffusion prior.
Because of this, the model cannot rely on a pretrained denoising prior.

**LoRA is not fine-tuning an existing diffusion prior; rather, it is learning a new clean-image manifold induced by our redefined endpoint.**

This is a conceptual shift from diffusion-based restoration.

## ▌2. Decoupling degraded content from the trajectory (no denoising-style refinement)

The degraded latent $E(I_{\mathrm{lq}})$ never enters the generative trajectory:
$$
z_t = (1-t) z_0 + t \epsilon,
$$
meaning the model is not “denoising” the degraded input.

It must instead learn to map noise to a clean target purely through our conditional RFM objective.
**This makes LoRA training fundamentally different from standard diffusion finetuning**, highlighting its non-incremental nature.

## ▌3. Why ITL + DLG are structurally necessary, not engineering tricks

> To illustrate why the three components interact structurally rather than additively, the conditional RFM objective can be written as:

$$
\mathcal{L}_{\text{RFM}} = \mathbb{E}\_{z_0,\epsilon,t} \Big[  \\|   v\_\theta ( z\_t, t, e\_f, [x\_t;  E(I\_{lq})])  -  (\epsilon-E(\mathcal{D}^{-1}\_\phi(I\_\text{lq})) \\|_2^2 \Big].
$$

Because our formulation discards the diffusion denoising prior, these components are not interchangeable but **structurally required**:

- **RFM** defines a stable deterministic transport from pure noise $\epsilon$ to the restoration-specific endpoint $z_0 = E(D^{-1}_\phi I\_{lq} )$ , which replaces the clean-image prior normally inherited from diffusion.
- **ITL** injects structural evidence through $h_t = [x_t; E(I_{\mathrm{lq}})]$, while keeping degraded statistics out of the trajectory and preserving fixed-length attention.
- **DLG** supplies the semantic priors needed to learn $\mathcal{D}^{-1}\_{\phi}$ from scratch: the system prompt provides a **task-level semantic prior**, and the image embedding provides an **instance-level semantic prior** for $I_{\mathrm{lq}}$.

Together, these components form the minimal conditioning structure compatible with our redefined RFM endpoint.

## ▌4. Practical scalability without diffusion priors (1B parameters)

Despite discarding diffusion priors, our reformulated RFM enables:

-  **1B trainable parameters** to unify SR, denoising, and colorization
-  **State-of-the-art or second-best results on 6 out of 7 benchmarks**, matching or surpassing **2.3B–2.4B diffusion-based baselines**, despite using only 1B trainable parameters
-  QHD native inference and **12K restoration capability** (first full restoration of the 12K Qingming scroll)

That LoRA can successfully learn such a complex inverse mapping **without leveraging diffusion priors** is a strong indicator that the method represents a conceptual shift rather than an incremental modification.

Furthermore, achieving these results with only **1B trainable parameters**, to our knowledge, the smallest parameter count among recent diffusion-based restoration models reaching comparable fidelity, underscores the efficiency of the proposed formulation. Achieving this level of performance **without** diffusion denoising priors highlights both the efficiency and the underlying conceptual strength of our approach.

## ▌Summary

Our method does not refine or utilize diffusion priors.

By redefining the RFM endpoint and restructuring the trajectory, we require LoRA to learn a new clean-image prior from scratch. This makes our approach **fundamentally different from diffusion restoration** and directly addresses any concerns regarding incremental contribution.

These clarifications may help contextualize several concerns that were based on assumptions specific to diffusion-style restoration, which do not apply to our redefined RFM formulation.

---

### Meta-Review · Area_Chair_Uy47 · 2025-12-25

**Summary:**

This submission proposes “In-Token Learning” for high-fidelity image restoration using a diffusion-transformer backbone and rectified flow matching, aiming to improve robustness under severe degradations and enable ultra-high-resolution inference. Review discussion revealed a major split: three reviewers argued the work is largely an incremental system combining existing components (RFM + channel-wise concatenation + multimodal conditioning), with overstated novelty and weak real-world generalization (notably on RealLQ250), while one reviewer viewed the paradigm shift as significant. The rebuttal emphasizes a redefined RFM endpoint and a pure-noise trajectory that purportedly removes diffusion priors, but the disagreement centers on whether this constitutes a fundamentally new method versus rebranding of known conditioning strategies, and whether the empirical evidence convincingly demonstrates broad real-world robustness.


While the paper is ambitious and reports strong results on several benchmarks, the committee does not yet have sufficient confidence that the contribution is fundamentally new rather than a system-level combination with re-framed terminology, and the real-world generalization concerns (e.g., RealLQ250) remain insufficiently resolved for acceptance. The review scores are highly polarized (8 vs 2×3). In this context, the single high score from wYA2 may be an outlier that places relatively more weight on a favorable interpretation of novelty than the rest of the panel and the current evidence support. Given the unresolved disagreement on the core methodological interpretation and remaining generalization doubts, I recommend rejection.

**Reviewer Concerns:**

- The authors provided an extended clarification arguing the method does not rely on diffusion denoising priors and instead uses a pure-noise trajectory to an inverse-degradation endpoint, and explained why ITL/DLG are “structurally necessary” under this formulation.

- Novelty / over-packaging: A key reviewer argues the central pieces are standard channel-wise concatenation and standard multimodal conditioning injection, and that the paper’s new terminology obscures limited methodological novelty. Even with the authors’ clarification, the core contribution can still be interpreted as a system-level combination rather than a new principle.

- Real-world generalization: A reviewer highlights that the method underperforms established baselines on RealLQ250 and considers the “conservative” explanation unconvincing and potentially indicative of overfitting to the synthetic degradation model.

- Ablation interpretation / causal evidence: Concerns remain about whether the reported ablations cleanly support the claimed causal mechanism (e.g., the relationship between “denoise-α” baselines and ITL conditioning).

- Score divergence due to interpretation: The authors state that the low scores are driven by a core methodological misinterpretation (8 vs 2×3 split), but this was not resolved through reviewer engagement, leaving the committee with substantial uncertainty.

**Reviewer Scores:**

Given the persistent disagreement and lack of convergence in discussion, I do not expect substantial post-discussion score movement. The current record reflects a polarized evaluation: one high score versus three low scores (2-level rejections), with the low-score concerns remaining outstanding (novelty and real-world generalization). The most plausible scenario is that the three low-score reviewers would keep their scores; the high-score reviewer would remain positive.

---

### Decision · Program_Chairs · 2026-01-26

Reject